



# Atmospheric aging of small-scale wood combustion emissions (model MECHA 1.0) – is it possible to distinguish causal effects from non-causal associations?

Ville Leinonen[1], Petri Tiitta[2], Olli Sippula[2,3], Hendryk Czech[2,a], Ari Leskinen[4,1], Juha Karvanen[5], Sini Isokääntä[1], and Santtu Mikkonen[1,2]

[1]Department of Applied Physics, University of Eastern Finland, Kuopio, Finland
[2]Department of Environmental and Biological Sciences, University of Eastern Finland, Kuopio, Finland
[3]Department of Chemistry, University of Eastern Finland, P.O. Box 111, 80101 Joensuu, Finland
[4]Finnish Meteorological Institute, Kuopio, Finland
[5]Department of Mathematics and Statistics, University of Jyvaskyla, Jyvaskyla, Finland

[a]now at: Cooperation group "Comprehensive Molecular Analytics (CMA)", Helmholtz Zentrum München, 81379 München, Germany

*Correspondence to*: Ville Leinonen (ville.j.leinonen@uef.fi)

**Abstract**

Primary emissions of wood combustion are complex mixtures of hundreds or even over a thousand compounds, which pass through a series of chemical reactions and physical transformation processes in the atmosphere ("aging"). This aging process depends on atmospheric conditions, such as concentration of atmospheric oxidizing agents (OH radical, ozone and nitrate radicals), humidity and solar radiation, and is known to strongly affect the characteristics of atmospheric aerosols. However, there are only few models that are able to represent the aging of emissions during its lifetime in the atmosphere.

In this work, we implemented a model (Model for aging of Emissions in environmental CHAmber, MECHA v 1.0) to describe the evolution by differential equation system. The model performance was first evaluated using two different, simulated datasets. The purpose of the evaluation was to investigate the ability of the model to 1) find the correct relationships between the variables in the dataset and 2) to evaluate the accuracy of the model to reproduce the evolution of variables in time. Subsequently, the model was implemented to wood combustion exhaust in atmosphere, based on a dataset from smog chamber
experiments. Evaluation in simulated datasets served as a basis of the drawings made from modeled aging of the residential wood combustion emission.

We found that the model was able to reproduce the evolution of the variables in time reasonably well. By using the state of the art detection algorithms for causal structures, we could unveil a large number of relationships for measured variables. However, as the emission data is complex in its nature due to multiple processes interacting with each other, for many relationships it
was not possible to say if there was a causal pathway or if the variables were just covarying.



This study serves as the first step towards a comprehensive model for the description of the evolution of the whole emission in both gas- and particle phase during atmospheric aging. We present contributions to challenges faced in this kind of modeling and discuss the possible improvements and expected importance of those for the model.

## 1 Introduction

Small-scale wood combustion is a common method for residential heating and has been identified as a substantial contributor to ambient levels of particulate matter (PM) in several European areas (Cordell et al., 2016; Glasius et al., 2018; Hovorka et al., 2015; Qadir et al., 2014; Reche et al., 2012). Wood combustion-derived aerosol, in particular from manually-fired logwood stoves, contains substantial amounts of several air pollutants, such as Black Carbon (BC), polycyclic aromatic hydrocarbons (PAH), volatile organic compounds (VOC) and CO (Orasche et al., 2012; Tissari et al., 2008), with consequences on Earth's

radiative forcing (Frey et al., 2014), cloud formation (Dusek et al., 2011), and human health (Kocbach Bølling et al., 2009). After release into the atmosphere, wood combustion emission are immediately transformed ("atmospheric aging") in a complex process involving multiphase chemistry, leading to oxidation and functionalization of particulate and gaseous pollutants (Hallquist et al., 2009; Pöschl, 2005) and consequent secondary organic and inorganic aerosol (SOA and SIA) formation. The formation of SOA has been reported to double to triple the concentration of the particulate organic aerosol emitted by wood

combustion, after less than 1.5 days of photochemical aging (Bertrand et al., 2017; Bruns et al., 2015; Tiitta et al., 2016). Despite several known reaction pathways for single emission constituents or compounds classes, such as PAH (Keyte et al., 2013), interactions between all constituents of a real-world combustion aerosol complicates the prediction of secondary aerosol formation and its physico-chemical properties.

Smog chambers or environmental chambers have been used to study the atmospheric transformation of single VOC or

combustion emissions for several decades, offering a controlled environment and conditions similar to the atmosphere, to study the effects of e.g. photochemical age, relative humidity, NOx, addition of seed aerosol, oxidizing agents etc. on emission transformation (Hinks et al., 2018; Lambe et al., 2015; Li et al., 2016; Platt et al., 2013; Tiitta et al., 2016). Previous studies have demonstrated that atmospheric aging alters toxicological effects of wood combustion aerosols (Künzi et al., 2013; Nordin et al., 2015) and changes the optical properties as well (Kumar et al., 2018; Martinsson et al., 2015), with implications for

climate and human health.

Radicals, such as OH, O3 and nitrate (NO3) are known to have an important role in SOA chemistry (Hallquist et al., 2009). OH is the most important radical during photochemical aging, whereas NO3 and O3 are more important during dark aging (Atkinson, 2000; Tiitta et al., 2016). SOA from photochemical and dark aging differs by chemical composition. However, both aging mechanisms are important and needed to take into account when evaluating the whole climate implications of aging

processes (Vakkari et al., 2014). Nitrate radical chemistry is an efficient SOA formation mechanism and also an important pathway for the production of organic nitrates, serving as a NOx sink in the atmosphere (Kiendler-Scharr et al., 2016).





Only few attempts have been made to capture the evolution of emissions in both, the gas- and particle-phase. Modeling approaches of SOA formation and evolution of the compounds leading to SOA can be divided to at least two groups. Some models, such as Volatility basis set (VBS), aim to describe one or several features of emission. In the VBS approach (Donahue et al., 2006) models the evolution is based on the volatilities of the compounds, considering the equilibrium concentrations of

different compounds in gas and particle phase, and how different factors, such as chemical and physical reactions affect the equilibrium state. Another approach to model SOA is the family of explicit chemical modeling. There exists several chemical models such as Master Chemical Mechanism (MCM) (Jenkin et al., 1997; Saunders et al., 2003) and GECKO-A (Aumont et al., 2005), which comprise large amounts of chemical reactions and pre-determined reaction coefficients to replicate the evolution of the system. MCM has been recently applied to wood burning emissions by running the model with most important

primary emission species to model the evolution of gas-phase species using smaller selection of reactions from the whole system (Coggon et al., 2019). Statistical Oxidation Model (SOM) is somewhere between one-quality models and explicit chemical models. SOM uses several qualities (volatility, number of carbon and oxygen) of compounds to predict SOA mass and atomic O:C ratio (Cappa and Wilson, 2012).

All approaches, volatility-based, SOM, and explicit chemical modeling, are based on differential equation approaches. These

equations describe the evolution of some quantity which we are interested to model and what can be transformed into the end-product of the model, here the amount of SOA. For the volatility approach this is the volatility distribution of particles, in chemical modeling the amounts of substances in the system.

In this kind of system, the changes of gases and particles in time can be thought to be a consequence of reactions occurring in the system. Reactions are caused by the initial compounds and certain properties of the compounds. Reaction coefficients and

concentrations of initial compounds determine the amount of reactions occurring in time. Compounds can be thought to be causally related as those are related to each other through chemical reactions. Initial compounds cause the increase in products and decrease in their own concentrations.

In this study, we described the connections between primary emissions and ageing conditions by using a causal model (Pearl, 2009). The aim of this study was to evaluate whether it is possible to learn the causal relations of variables from the system of

atmospheric aging without having explicit prior information about these relations. On that account, we created a model for the complex interactions between aerosol constituents in both gas and particle phase (Hartikainen et al., 2018; Tiitta et al., 2016). Here, we present the first version of our model that is able to represent dependencies between observed variables on the chamber studies in (Tiitta et al., 2016). In addition, we discuss the issues in data processing related to causal modeling in wood combustion aging, and more generally, emission aging datasets by applying the model to artificial, simulated datasets and

evaluating the accuracy of the model.



## 2 Data and methods

The chamber experiments to study atmospheric transformation of residential wood combustion emissions were conducted in the ILMARI environmental chamber (Leskinen et al., 2015) at the University of Eastern Finland, as described by Tiitta et al. (2016). Briefly, five chamber experiments were conducted using a modern heat-storing masonry heater (Reda et al., 2015) as
the emission source. The masonry heater was operated with spruce logs, using both fast ignition and slow ignition for initiating the combustion experiment (Detailed procedure described in Tiitta et al., 2016) to adjust VOC-to-NOx ratio. In each experiment, a partial sample of the combustion exhaust was diluted and fed into the chamber for 35 min, including the ignition, flaming combustion and residual char burning phases of logwood combustion (Czech et al., 2016). This was followed by stabilization of the sample, after which the oxidative aging of the sample was initiated by feeding ozone into the chamber.
Both, dark- and UV-light aging experiments were conducted, representing evolution at night- and daytime in the atmosphere, respectively.

We used data from two different types of experiments: In the first one, UV-lights (blacklight lamps, 350 nm) were switched on immediately after feeding ozone into the chamber, and the wood combustion emission was photochemically aged for four hours (Experiments 4B and 5B, Tiitta et al., 2016), which is based on the measured OH-radical exposure represented 0.6-0.8
days of photochemical aging in the atmosphere. These experiments simulate daytime ambient conditions where OH-radicals are dominating the oxidative aging of emissions. In the second type of experiments, aging was conducted at first without UV radiation and after 4 hours of dark aging, the UV lights switched on (Experiments 2B and 3B, Tiitta et al., 2016). The dark aging period represents night-time ambient conditions where ozone and nitrate radicals are dominating the oxidative aging of emissions (Brown and Stutz, 2012; Tiitta et al., 2016).
The evolution of gases and particles in the chamber was measured by comprehensive on-line measurements. Gas-phase organic and inorganic compounds were measured by single gas analyzers and a Proton-Transfer-Reactor Time-of-Flight Mass Spectrometer (PTR-TOF 8000, Ionicon Analytik, Innsbruck, Austria). The PTR dataset have been described in detail by Hartikainen et al., (2018). The particle size distribution was measured by a Scanning Mobility Particle Sizer (SMPS, CPC 3022, TSI) and the chemical composition and mass concentrations by a Soot Particle Aerosol Mass Spectrometer (SP-HR-
TOF-AMS; Onasch et al., (2012)).

Dark and UV-induced aging were treated separately in further data analysis because intensive UV radiation affects the transformation of emission due to photochemical reactions and further the dependence structure of variables. The aging type can affect, for example, amount of SOA formed, reactions of OH and gaseous nitrous compounds, and their formation products (Ortega et al., 2013; Vakkari et al., 2014). We had two experiments with different ignition technique, one with fast ignition
and one with slow ignition for both dark (2B and 3B) and UV-light induced aging (4B and 5B). Both experiments with the same aging type were used as a single dataset for the model.





Ignition type affects the composition of the emission from wood combustion (Hartikainen et al., 2018; Tiitta et al., 2016). As we used both experiments with different ignition types as a single dataset, we assumed that the reactions in the chamber are similar in both datasets and only the concentrations of the compounds are different.

### 2.1 Data processing

Here we present step by step how data has been processed before modeling. In this study, the data processing consists three steps that are listed here and described in more detail in Sect. 2.1.1-2.1.4:

1. Using dimension reduction methods for the particle size distribution and the mass spectrometer datasets.
2. Forming approximate time series for the OH based on measurement of butanol-d9.
3. Adjusting the time resolution of data to be the same in datasets from every measurement instrument.

4. Filtering time series to reduce the amount of measurement error.

### 2.1.1 Dimension reduction of SMPS, PTR, and AMS datasets

SMPS provided very detailed measurements of size distribution consisting of over 100 size bins by particle (mobility) diameter from 14.1 – 14.6 nm, to 710.5 – 736.5 nm. For this study, we used a simpler representation of size distribution, size binning of wall-loss corrected (see Tiitta et al., (2016)) SMPS time series which were grouped into four larger size classes. These four

classes roughly represent atmospherically relevant particle modes: under 25 nm (nucleation mode), 25 to 100 nm (Aitken mode), 100 to 300 nm (accumulation mode), and over 300 nm (coarse mode). Measured size bins were summed to form number concentrations of particles in four classes.

Mass spectra of AMS and PTR contain over a hundred variables each. In order to form a model that provides a simple representation of emission's evolution, we do not want to present the dependencies of all single ions of mass spectra. Instead,

we try to represent the evolution of spectra by combining masses into a smaller number of variables, called factors, by applying dimension reduction techniques for both gas- and particle phase measurements from PTR and AMS, respectively. Compounds in the same factor are assumed to behave similarly, as the factors are formed from using dependencies in the dataset. However, it does not mean that all compounds in the same factor will react with the same compounds such as oxidizers. That can make interpretation of factors in the model more difficult.

Unfortunately, it is not necessarily self-evident which dimension reduction method should be applied to the dataset of interest. Isokääntä et al. (2019) have studied the importance of selecting the appropriate dimension reduction method to the dataset in case of atmospheric measurement datasets. Datasets were produced by mass spectrometers in chamber studies. They found that fragmentation of compounds and rapid changes in chemical composition (e.g. caused by turning on UV-lights) should be taken into account when selecting the dimension reduction technique. They suggest that for datasets, where fragmentation is

not such a problem, exploratory factor analysis (EFA) or principal component analysis (PCA) might extract those fast changes from the data more efficiently than e.g. positive matrix factorization (PMF).



Our aim was to compare multiple experiments in further data analysis, and therefore dimension reduction techniques were applied to the PTR datasets from different experiments at once. Using all of the data to form the factors instead of applying a dimension reduction method separately to each one of the experiment have some advantages. Same variables (masses) are loaded to the same factor in each experiment, which means the comparison of factors is easier between experiments. However,

as experiments contain different burning and aging conditions, proportions of masses loaded into one factor can be different between experiments. For example, a different type of ignition might produce a large amount of specific compound that is absent in other experiments.

For PTR-MS mass spectra, EFA and PCA were tested, but EFA was selected further analysis due to more interpretable factors. EFA was applied using the function *fa* from package *psych* (Revelle, 2018) in the R environment (R Core Team, 2019).

Minimum residual technique was used for EFA and oblique Oblimin-rotation was used to enhance the separation of the masses. In addition, masses that had no significant contribution (i.e. loading) on any of the factors have been removed from analysis and factorization was performed again to the remaining masses. These removed compounds were mainly instrumental background or compounds with very small concentration without any clear structure as a time series. Time series for the factors were calculated by sum score method, where the original data is multiplied directly with the loading values from EFA with

possibly some threshold limit (Comrey, 1973). We selected absolute value of 0.3 as a threshold for the loadings, meaning any loading values smaller than that were suppressed to zero before the multiplication (Yong and Pearce, 2013). PTR factor 2 was scaled to be positive by adding the minimum value to all the time points. Negative values were caused by the baseline correction of PTR data and factorization. The factors (see Fig. 1) were identified based on their temporal evolution and the compounds as well as their average carbon oxidation state ($OS_C$) contributing in each factor, and for PTR, three factors were found: 1)

primary VOCs, 2) photochemical aging products, and 3) dark aging products.

Factorization of AMS spectra by PMF has been described in Tiitta et al., (2016). OA spectra were analyzed applying positive matrix factorization (PMF; (Paatero, 1997; Ulbrich et al., 2009)) to classify OA into subgroups according to their formation mechanisms. They found two factors to describe the primary organic aerosol particles (POA): 1) biomass burning OA and 2) hydrocarbon-like OA. In addition, they found secondary organic aerosol (SOA) factors representing the three major oxidizers:

1) formation by ozonolysis, 2) formation by nitrate/peroxy radicals and 3) formation by OH radicals. PMF factorization was performed using The PMF Evaluation Tool v.2.08 (Ulbrich et al., 2009).

We assume that the factors from AMS and PTR dataset represent the groups of products we have determined based on representative compounds and their evolution. Exact representation of the evolution of specific groups of compounds would require knowing the exact compounds in the chamber, which is not possible with current measurement setup (Canagaratna et

al., 2007; de Gouw and Warneke, 2007; Hatch et al., 2017; Yuan et al., 2017). Accurate optimization of factors is out of scope of this modeling study.



### 2.1.2 Forming OH time series

As OH radical has an important role in transformation of emission during photochemical aging, we applied OH concentration level estimation formulas (1-3) from (Barmet et al., 2012) to calculate the OH concentration time series. Its estimation was based on the d9-butanol tracer method, and the slope was determined from 20 observations (time period of 30 min) around the time point OH concentration was estimated. Concentrations of OH, which were estimated to be negative, were set to zero. Negative values were only estimated for dark aging experiments, however, in according to our initial assumption, OH was not allowed to affect other variables.

### 2.1.3 Adjusting time resolution

The applied instruments measured concentrations and size distributions with different time resolutions. PTR-MS measured every two second, whereas SMPS scan time and sample analysis took a total of 5 minutes to produce a single measurement. To combine variables measured with multiple measurement devices into one dataset, it is important to transform data appropriately to make variables to the same time resolution. We applied simple cubic spline interpolation to interpolate SMPS data to same time resolution than AMS data, which is 2 minutes. Data from PTR-MS and other gas analyzers was averaged over a 2-minute time period.

### 2.1.4 Filtering time series

Measurements are discrete observations from continuous transformation process of emission and contain measurement error, which may arise from measurement limitations regarding the representativeness of measurement, accuracy of a measurement instrument or some other factor affecting to measurement event. The measurement $y_{k,t}$ at time $t$ for time series $y_k$ is the sum of the true value $\alpha_{k,t}$ and the measurement error $\varepsilon_{k,t}$. These measurement errors comprise sampling from the chamber ($\varepsilon_{k_{sampling},t}$), error related to the estimation of particles and gases losses to chamber walls ($\varepsilon_{k_{wall-loss}}$), and error related to processing of measurement instrument data from raw data into more useful form ($\varepsilon_{k_{processing}}$).

Measurement $y_{k,t}$ were presented as

$$y_{k,t} = \alpha_{k,t} + \varepsilon_{k,t} \quad (1)$$

$$\varepsilon_{k,t} = \varepsilon_{k_{sampling},t} + \varepsilon_{k_{wall-loss},t} + \varepsilon_{k_{processing},t} \quad (2)$$

where error term $\varepsilon_{k,t}$ is independent in time and follows specified distribution presenting all uncertainties. State $\alpha_{k,t}$ describes the estimate of the real state of the variable in the chamber and the error term $\varepsilon_{k,t}$ represents the error related to estimation of the state.

Understanding the evolution of the state $\alpha_{k,t}$ of the variable is the question of interest. We would like to understand the factors affecting the change of state during the aging of emission. Therefore, we estimated the state $\alpha_{k,t}$ from measurements $y_{k,t}$ for each variable.



Multiple statistical methods can be applied to estimate the state $\alpha_t$ from data. We applied the filtering method that used only already observed measurements to determine the current state of the time series. The method was similar to LOESS (locally estimated scatterplot smoothing (Cleveland, 1979)), where observations are weighted according to the proximity of measurement $y_{k,t1}$ from the measurement $y_{k,t0}$ which state is estimated. Instead of using all observations from the time series

(smoothing), we used only observations before the estimated state (filtering). We applied the method to every time series separately and the number of previous measurements used (time window) to estimate the current state $\alpha_{k,t0}$ was determined separately to each time series.

The filtering window was tuned manually by evaluating the state by comparing the time series and the trend and choosing window such that the trend is representative to time series. Because of different measurement instruments and time resolution,

different time series had a different fraction of noise to total variation. Figure 2 shows the effect of filtering for one variable during experiment 2B.

## 2.2 Causal model and causal structure

A causal model is a model that allows the researcher to analyze complex dependence structures between observed and latent variables. Here we refer to the definition of the causal model from Pearl, (2009) def. 7.1.1. The causal model defines functional

relationships between variables and thus determines the dependence structure between all variables.

The causal structure is a representation of dependencies between variables without explicitly determining the exact functional form of each dependency. It can be presented as a directed acyclic graph, where nodes represent variables. Edges connect nodes and represent dependencies between variables. The direction of an edge determines the causal direction of the dependency. Figure 3 shows an example of a causal graph.

The causal structure can represent general knowledge about the phenomenon and can be determined without using any data from experiments, or by applying causal discovery algorithms to search the structure or some part of the structure. In this study, we determined the causal structure between variables by using both previously known dependence paths between variables and causal discovery algorithms to discover previously unknown dependencies from measured data. Because of uncertainty in the structure found by causal discovery algorithm, the meaningfulness of each found dependency was evaluated.

## 2.3 Causal inference using causal model and causal graph

A important feature that separates the causal model from traditional multivariate model is its ability to make both interventional and counterfactual inference (Pearl, 2018). This enables the researcher to study questions such as how intervening of a variable changes the state of the observed process in the future (interventional), or how the currently observed state would have changed if we had changed some conditions beforehand (counterfactual). These questions are examples of interventional and

counterfactual questions. This kind of inference is possible because of the assumption that the causal model represents the

effects of variables to each other (Pearl, 2009). When using the causal model, the one assumes that the causal structure contains all observed and latent variables needed to explain the studied phenomenon.

Causal models lie between statistical and physical models (Peters et al., 2017). As physical models, such as global climate models and MCM model (Jenkin et al., 1997; Saunders et al., 2003)), causal models can also consider interventional and
counterfactual questions using simulations where the researcher can alter initial parameters or go back in time from the observed state of the system. These questions cannot be assessed using only statistical models. Causal models do not always provide the same physical insight of the phenomenon that physical models do. However, the calculation of simulations in physical model may be time-consuming if the model is complex. In this study, the causal model provides a simpler representation that does not use as many parameters as the physical model and thus is computationally more feasible.

**2.4 Causal discovery algorithms**

Causal discovery algorithms are known as algorithms that enable searching for dependence structures of given variables using data and observed dependencies between variables to eventually find the causal structure. These algorithms are usually based on (in)dependency tests, data fitness-based scores, or both in addition to prior knowledge about the structure to complete the causal structure. Independence tests are often based on correlation, which implies that the data is assumed to be multivariate
normally distributed and that the observed dependencies between variables are linear. Both normally distributed and linear dependency assumptions are often in place in score-based algorithms.

When determining the causal structure, it is possible to include initial knowledge in the structure. As mentioned above, algorithms are based on observed dependencies in dataset and distinction between causal and correlated dependency might be difficult. R-causal package (Wongchokprasitti, 2019) enables the researcher to forbid or to require edges previously known as
impossible or necessary to be in the structure, respectively. For example, previously known dependencies between variables can be added to the causal structure before applying the algorithm and forbid known, not causal directions of connections. Also forbidding of edges that are impossible because of other reasons, e.g. because of the temporal structure of variables, can be removed.

**2.5 Creating the model**

The main scope of this study is to understand the evolution of measured gases and particles. For describing the evolution of each variable, we devise differential equations. The change of measured variable/concentration between subsequent time points has been modeled using previously observed values of relevant variables as predictors explaining the change. Figure 4a) illustrates the way variables are linked in a causal graph. We assumed that only the previous observation/concentration measured right before time interval can affect differences in variables during the time period. We modeled the evolvement of
each variable using linear differential equations to model the evolution of variables, as defined in formula (1). For each variable of interest $x_j, j = 1, \ldots, n$, Eq. (3):


$$\Delta(x_{j,t}) = x_{j,t+1} - x_{j,t} = \beta_0 + \beta_1 tx_{1,t} + \beta_2 x_{2,t} + \cdots \quad (3)$$

describes how the difference of variable value between subsequent time points is determined by values of measured variables right before time interval. Predictors used to model $\Delta(x)$ are connected to the variable in the causal structure. In other words, the first part of the study is to find the predictors to $\Delta(x)$:s.

Below we present a simple four-step procedure of how the model was applied to processed version of emission dataset. Sections 2.5.1-2.5.4 provide more detailed description of selected steps in model creation.

1.      Use the causal discovery algorithm to search the potential dependencies between measured variables and measured changes ($\Delta(x):s$) in time. For each measured change, we get its own potential variables, which are possible causes of the measured change.

2.      Form interaction variables from measured variables such that both variables in the interaction variable should be suggested by the algorithm. In addition, prior assumptions have been taken into account here, thus forbidden variables cannot be part of the interaction variable. Interaction variables are also formed separately for each $\Delta(x)$.

3.      Use LASSO (least absolute shrinkage and selection operator) to select predictor variables amongst interaction variables (and possible single variables suggested by the algorithm). Simultaneously, estimate the coefficients of each selected

predictor using a linear model.

4.      Calculate the modeled evolution using ODE system *deSolve* (Soetaert et al., 2010) using estimated coefficients as reaction coefficients and the first observation from the experiment as initial state.

### 2.5.1 Using a causal discovery algorithm

Causal discovery algorithms implemented in r-causal package (Wongchokprasitti, 2019) enable to "tune" the amount of
dependencies in the whole graph. In PC-algorithm, tunable parameters are depth and alpha. According to the algorithms documentation (Wongchokprasitti, 2019), the depth determines 'a number of nodes conditioned on in the search'. Depth values used in the algorithm were two, three, and infinite, meaning that every node can be further conditioned in the search. Alpha, which is set between zero and one, indicates statistical significance of the dependency between searched variables. However, significance was used here as a tuning parameter to allow the number of edges to rise with larger alpha values. Alpha values
0.01, 0.05, 0.2, 0.3 have been used for wood combustion datasets. For simulated datasets alpha 0.05, 0.2, and 0.3 were used. For depth, 3 and unlimited depth (-1) was used.

Causal discovery algorithms have been used to search potential cause variables for each variable. An important part of the causal discovery algorithm is the implementation of prior knowledge. We used prior information for mostly restricting dependencies which were thought to be impossible or negligible. Separate prior information was used for dark aging and
photochemical aging. The apparent *a priori* difference in prior information between two aging types is that in dark aging, OH is not allowed to affect any of the other variables. In photochemical aging, OH can affect other variables than particle size distribution variables. OH was also forced to affect SOA3, which was characterized as OH radical formation products in a





previous study (Tiitta et al., 2016). Furthermore, effects from size distribution variables on other variables, POA on SOA, AMS variables on gas phase variables, and negligible variables (variables with small concentration) on any other variable were not allowed. As the applied causal discovery algorithm is based on correlations, variables that have negligible effects in reality, could have been found to have large effect in correlation-based search.

**2.5.2 Forming interaction variables**

Causal discovery algorithm provides a list of potential edges between variables which we wanted to turn into interaction variables. We used this list from the discovery algorithm as a starting point to form a final causal structure to use in a model. For each variable Y, the algorithm provides a list of variables X, that could cause Y, X➔ Y. We used those variables X to form interaction variables for explaining Y.

Interaction variables for Y were formed by pairing potential causes of Y. Pairs were formed so that both variables are suggested by the causal discovery algorithm. The only limitation for variables is that variable interactions are not allowed to violate prior assumptions. For example, if it is known that Z cannot cause Y, it cannot be included into interaction variables that could cause Y. In this step, we formed a list of potential interaction variables for each variable Y. Figure 4b) shows an example of interaction variables formed.

In addition to interaction variables, we allowed some single variables also to act as predictor variables. Many physical and chemical reactions involve more than just one compound or phase state, but some reactions involve only one variable. As an example, particles of the same size can coagulate during evolution and form larger particles.

Similar solutions appear in the literature. In chemical kinetics theory, one or more variables are reacting to form other compound(s). The difference of end compound concentration in time can then be represented as a product of concentrations

of reacting compounds multiplied by the rate coefficient and time $t$. Interaction variables and some single variables found by algorithm were used to select proper explanatory variables for the effects of variables to the response variable $\Delta(x)$.

The calculation of interaction variables might be done directly by using all possible variables in the dataset and the select the proper explanatory variables for each response variable amongst all interaction variables by causal discovery algorithm or some other method. However, if the amount of initial (single) variables is $n$ then the number of possible two-variable

interactions is $\binom{n}{2} = \frac{n!}{2!(n-2)!}$, which would be already large for relatively small $n$. Therefore, a pre-selection of most potential variables to form these interaction variables reduces the number of interaction variables formed remarkably.

**2.5.3 Applying LASSO to the dataset**

Interaction variables also brought some challenges for selection of variables. Interaction variables are often highly correlated, as two interaction variables can have one common variable. Highly correlated variables can lead to multicollinearity in

coefficient estimation. When data have multicollinearity problem, coefficients of interaction variables might suffer from





biases, and small changes in the model can lead to large changes in its output. Therefore, it is important to note the possibility of multicollinearity in predictors.

When interaction variables have been formed, there are usually many correlated variables that can explain each response variable $\Delta(x)$. We used LASSO regression (Bayesian approach from (Wang, 2019) based on (Park and Casella, 2008)) to

reduce the number of variables and multicollinearity in the set of explanatory variables. LASSO shrinks the coefficients of some redundant variables to be (almost or equal to) zero and therefore reduces the number of effective variables. Variables with coefficient close to zero were consequently left out from the structure. Variables that have non-zero coefficient in LASSO fit were used as predictor variables for $\Delta(x)$s. Coefficients of these variables were used as coefficients in the final model. Figure 4b) shows an example how some interaction variables are left out after using LASSO.

Negative coefficients are only allowed when predictor variable includes the variable that is predicted i.e the amount of predicted variable is decreased due to the process. Without this limitation, the modeled value of variable can go below zero during calculation, which is not physically realistic. If a variable that has negative coefficient does not have predicted variable in response variable, the model attempts to fix that by changing the predictor to some other predictor that will have positive coefficient (usually negatively correlated with original predictor) or to some predictor that includes the predicted variable.

### 2.5.4 Calculating the evolution based on obtained model

We evaluated the model fit by calculating the evolution of the system based on model's structure and estimated coefficients. We used R package *deSolve* (Soetaert et al., 2010) to calculate the evolution of modeled linear ordinary differential equation system.

Selection of the best model was then done based on that evolution. We calculated Root Mean Squared Error (RMSE) for each

variation of tuning parameter and selected the model based on smallest RMSE for calculated evolution.

As we had only two datasets for both dark and light induced aging, we did not want to use separate datasets for training and evaluating the model fit. As the ignition type also affects e.g. the chemical composition of the emission (Tiitta et al., 2016), evolution between datasets might not be directly comparable. Therefore, we used both ignition types for searching the structure and estimating the coefficients of the model.

### 2.6 Simulation studies

For the situation where the causal graph between variables is not completely known, causal discovery algorithms have been suggested to be one solution to find missing dependencies. The atmospheric aging process of wood combustion emissions is definitely one of these situations at present where we have limited prior knowledge. For the reliability of the obtained results from our experimental dataset, it is important to know how causal discovery algorithm combined with our model is functioning

in a similar kind of datasets than our experimental dataset (see Table 1).

Several questions of interest existed related to the properties of input dataset and data pretreatment (Table 2). Firstly, we were interested to study how the precision of the measurements by analytical instrumentation is related to the model fit, which was





assessed the measurement error and the number of measurements made per time bin. In order to mimic the measurement error, normally distributed random noise was added to the variables.

Secondly, we were interested to know whether the methods we applied to increase the quality of data are actually increasing the quality, accuracy of fit and structure of the model. Does filtering or smoothing of time series improve the fit and accuracy

of prediction and is there an optimal time resolution to which data should be averaged?

Thirdly, the amount of necessary prior information was the question of interest. We were interested to study the importance of prior information to the modeled structure. Does addition of prior information improve the accuracy of modeled structure and how much incorrect edges do we have? Additionally, the effect of prior information to the model fit was interesting. How much prior information is necessary to get a reasonably good fit for the model?

The question about necessity of the prior information is also related to the dependence of model fit and the correctness of the structure. Intuitively, one might think that the correct structure would produce the best fit for the evolution. As many of the variables are highly dependent, it is probable that we will fail to obtain the real structure between variables. If dependent variables we use in the model to explain the evolution are correlated with real causes in the dataset, we might still be able to predict the evolution of emission in the chamber. In addition to the differences between obtained and real structure in the

model, we are interested about the predictive value of the obtained model compared to the simulated evolution.

Two simulated datasets (see Table 1) were formed to study how model would perform in a situation where we know the correct evolution and structure. Both dataset were formed using R-package deSolve (Soetaert et al., 2010). Datasets describe the evolution of Ordinary Differential Equation (ODE) system which length is 100 time points. The difference between datasets is the way how differential equations of variables are linked to each other. In smaller datasets, differential equations are from

Mass Action Kinetics system. In larger dataset, equations have been formed independently for each variable. We called these datasets as simulated datasets throughout the text.

Accuracy tests for causal discovery algorithms were also performed earlier (Scheines and Ramsey, 2016; Singh et al., 2017). However, those tests are dependent on the used dataset. In our case, variables that explain the evolution of some variable do not originate directly from the discovery algorithm. Instead, the variables from the algorithm are only used to form possible

interaction variables, hence this situation differs from the tests performed earlier.

We measured performance of the model in simulated dataset by two ways. First is the fitting of the model to the simulated dataset: how well the model can capture simulated evolution and how well the model can predict the simulated evolution after fitted dataset (ability to predict). Second can be called as structural accuracy: how well the underlying causal structure of variables can be returned by the model.

For measuring accuracy of the model fit, we compared the evolution obtained from the model to measurements. Evolution was then compared to true evolution, not including the error added to the simulations, using RMSE for all time series. To equally weight each time series when calculating RMSE, each time series were scaled by dividing those with a standard deviation before calculating RMSE. In further text, we refer to this scaled version as RMSE.





In addition to the accuracy of the model, the accuracy of model prediction was evaluated. We used the obtained coefficients from the model to predict further time steps of the evolution of the system. Then we compared the prediction to the same time steps from the real system and evaluate the accuracy of model prediction using RMSE. Prediction length was 30% of the simulated dataset.

As mentioned before, RMSE calculated from the used dataset was compared to the real RMSE, which was calculated comparing evolution from the model to the error-free evolution from the simulations. In reality, we obviously do not have an error-free dataset. Hence, RMSE is calculated from dataset containing measurement errors. The purpose of this investigation was to see whether RMSE that is calculated from error-free dataset is in line with the metric that is calculated from dataset containing measurement errors. The reason why these two estimators might give a different result is related to modifications

of the data before applying the model or biases in errors of the obtained dataset. In this case, the interest was to see whether applied filtering and smoothing techniques produce bias to the dataset and therefore goodness-of-fit measures would have significantly higher values in the error-free dataset than what the values should be based on the obtained dataset.

For structural accuracy, we used adjacency precision (AP), adjacency recall (AR) (Scheines and Ramsey, 2016) and F-score (Singh et al., 2017). AP was defined as a fraction of correct edges in the model of all proposed edges. AR was defined as a

fraction of correct edges in the model of all correct edges. F-score was defined based on AP and AR as

$$F_{score} = 2 * \frac{AP*AR}{AP+AR}.$$

In addition to F-score, we wanted to study whether incorrect predictors for variables were close to correct causes and if the model is able to find a good replacement for each correct predictor that was not chosen for modeled structure. Correlation was used to measure closeness here. For each correct predictor we calculated correlation between it and each predictor in the model

(for same variable). The maximum of these correlations was taken as the value for that correct predictor. CorMean is the mean value of correlations for all the correct predictors.

## 3 Results and discussion

### 3.1 Simulation studies

The effect of measurement error is reported for simulated datasets (Tables 3 and 4). We found that in general, the model fit to

the data was better when the error was lower. This is not surprising, as the error makes the dataset noisier. The effect of uncertainty was small for F-score and correlation between the correct edges and edges in the model. Furthermore, in both datasets the optimum number of measurements during one experiment was related to the amount of error applied (Tables 3 and 4). For lower error, higher time resolution leads to better results when the real change between subsequent time points was dominating the change observed and if many observations about the evolution process were available. In case of high error,

fewer observations from the phenomenon were preferred, as the signal to noise ratio was low and increasing the number of observations would make it even lower. However, the dependence between sample size and fraction of error was weak.



Filtering or smoothing of only time series representing measurements turned out to reduce the error when measurement uncertainty was high (Table 5). These methods were also improving the accuracy of model prediction. Again, the importance of applying smoothing or filtering was larger when the measurement uncertainty increases.

To understand importance of applying smoothing or filtering method in a real dataset, we need to understand whether the error
in the real dataset is large enough that applying filtering or smoothing method is reasonable. We assume the error related to the time series as the difference between these datasets. The standard deviation of the error was then compared to the standard deviation of the filtered dataset to estimate the fraction of the error of the whole standard deviation in the time series.

In AMS and PTR datasets and in largest size bin of SMPS measurements, the fraction of standard deviation of error was around 10-20% of the whole standard deviation. Variables measured by gas analyzers contained the least amount of error,
approximately 1-5% of the standard deviation of filtered time series. However, because the fraction of error of some variables is high, applying of the filtering methods to real dataset was reasonable.

Reasons why applying the filtering methods to the dataset is not improving the fit before the error is relatively high can be assessed further, but was omitted in this study. In addition to the fraction of error, the sample size is probably related to the necessity of the filtering method. These simulations for the larger dataset were performed using the sample size of 100, which
is the same sample size that was used in the shortest experiment.

Fourthly, prior information was closely related especially to the correctness of the obtained dependence structure between variables in the dataset. In some situations, there might be large amounts of prior information from previous studies available, which might help to construct the structure based on prior knowledge. This means that the correctness of the structure is high, and the causal inference based on the structure and the observed dataset has higher quality.

From wood combustion emission experiments, we do not have much prior knowledge about the phenomenon. Even though there has been and currently is a vast amount of research about the oxidative aging of combustion emissions, the number of possible reactions occurring during experiments is very high and the reaction coefficients are therefore hard to estimate. Thus, the construction of the "true" causal structure between variables might be out of reach based on low amount of prior information.

One question of interest is that how the incorrect edges in the structure are related to the missing or present correct edges. Based on the nature of the causal discovery algorithm one might think that the algorithm is not able to separate between causal and non-causal associations. Therefore, it might be that in the model the real cause is replaced with the indicative one. This, of course, decreases the accuracy of the structure, but not necessarily affect the fit and accuracy of prediction of the model, if the dependence between a real and an indicative cause is strong enough. Decreased accuracy in the structure is obviously
causing the incorrect interpretations of the causal effects.

We found that there is a small difference in both fittingness and accuracy parameters when the amount of correct and incorrect edges in prior information is varied in the smaller, artificial dataset. Larger amount of correct prior information led to slightly lower RMSE for the model and prediction (Table 6). Addition of both correct and prior information led to lower RMSE and higher F-score. For prediction, the information had not much effect.





## 3.2 Modeling results for combustion experiments

As the evolution of emission is known to be different in dark and UV-aging conditions, experiments are separated based on aging type. In addition, prior knowledge for dark and UV aging experiments are of different quality (see Sect. 2.5.1).

The causal discovery algorithm was used for both dark aging experiments (dark aging part of 2B and 3B) using both
experiments as input dataset for the model. Similarly, UV aging part of 4B and 5B was used to search the structure for UV aging. Thus, both experiments with same aging type have the same dependence structure between variables and also the coefficients for the experiments of same aging type were the same. The dataset used to estimate the coefficients included the whole data from both experiments. After estimating the coefficients, modeled evolution of the system was then calculated and compared to the measured evolution of the experiment. The zero-time in each experiment is the start of the aging period of the
experiment.

For dark aging experiments, the causal graph for SOA variables is shown in Fig. 5. Coefficients for the edges are provided in the Supplement. For SOA, most of the edges connected seem to be possible causes. For SOA1, the main increase of the particle concentration is due to the reaction with ozone. According to the model, some parts of the SOA1 might turn into SOA2 due to reactions with nitrate radical. For SOA2, reactions of nitrogen species ($NO_2$ in gas-phase) are involved in most of the reactions.
For SOA3, the change in concentration is minor compared to SOA2 during dark aging. According to the model, some increase in SOA2 and SOA3 is occurring due to ozone reactions.

We see from Fig. 6 that the evolution based on the model for both experiments follows the measured evolution well. In general, some factors (SOA1, SOA3, and PTR2) have slight differences between measured and modeled evolution in either one or both experiments. All coefficients for dark aging model and figures for other variables can be found in Supplementary file (Table
S1 and Fig. S1-S3).

For photochemical aging experiments, the difference between measured and modeled evolution is still slightly larger than for dark aging experiments. The largest differences between measured and modeled evolution were in variables NO and $NO_2$ on both experiments.

Figure 7 shows the causal graph for SOA variables in photochemical aging experiments. For SOA3, which has the most
significant increase during the experiments, reactions of OH with PTR1 and photochemical aging products reacting with SOA3 are the most important sources. SOA1 is also related to OH during photochemical aging, whereas SOA2 is mostly related to reactions with gas-phase products (PTR2). Figure 8 shows the evolution of PTR3 and SOA3. All coefficients for photochemical aging model and figures for other variables can be found in Supplementary file (Table S2 and Fig. S4-S6).

## 4 Summary and conclusions

In this study, we created a model for describing the evolution of complex system based on differential equations. Model creation is described step-by-step in Sect. 2.5. Dependences between observed time series were studied using causal discovery algorithm. We tested the accuracy of the fit, prediction and structure returned model using simulated datasets. Predictive



accuracy of the model can be considered a reasonable measure for many possible applications of the model, where explicit causal understanding of the system is not needed.

We have also assessed many of the uncertainties related to atmospheric measurements and studied how these affect our model in simulated datasets (step-by-step description of data processing is provided in Sect. 2.1). Based on simulations, we managed

to reduce the error related to the measurement process by smoothing the initial time series before applying the model, and therefore make the model and the obtained structure more accurate. These errors are important to be considered in similar studies also in the future.

It is evident that the modeling of the structure and real causal paths of the evolution of combustion emission would need more initial knowledge about the dependencies between different gaseous and particulate matter. The model can be applied to test

if hypothesized paths can be seen in the data, though there are still many possible paths that cannot be distinguished from these data that can lead to the almost same result at the end of the experiment. However, chamber experiments make it possible to study the effect of each variable to another and reducing the amount of correlated but not correct variables to explain the variables of interest. This can be achieved by controlling the number of compounds so that only one of the possible explaining variables are able to affect the variable of interest. Then it is possible to measure whether supposed interaction between

compounds/variables is present or not. Also increasing the number of experiments in the model would probably lead to a better result, as some of the dependencies measured in some experiment can be disapproved in other experiments if those are not causal.

For wood combustion emission experiments, the modeled evolution was very close to the evolution measured in the chamber. Some components of the experiments, such as ignition type and the resulting composition of the emission, can affect the

composition of the PTR and AMS factors. The composition of those factors can result the coefficients being different in experiments. Therefore, model described the evolution of experiments well.

However, it must be considered that our model is not based on the explicit knowledge of chemical formulas of the compounds, or the number of carbon and oxygen of the species. In the case of complex systems, such as wood combustion, which contain hundreds to thousands of different emission species, explicit knowledge is not available. The presented model used temporal

evolution of the compounds in 1) factors made for particle- and gas-phase mass-spectrometer data and 2) forming a structure of dependencies between compound groups and some specific compounds. In that sense, the model differs from SOM and MCM (Cappa and Wilson, 2012; Jenkin et al., 1997; Saunders et al., 2003).

Based on simulated experiments we made, it seems that the model obtained from the procedure we have explained here is not strictly a causal model, as the procedure is unable to find the correct edges among all possible edges without excessive

knowledge about the phenomenon. Therefore, the results of the model cannot directly be interpreted as causal evolution of the combustion emission in the atmospheric chamber. However, it seems that the model can capture the evolution quite well, and most of the incorrect edges in the simulated model are correlated with at least some of the correct edges and can most probably be interpreted as indicators for edges not measured in data but affecting the evolution. Based on simulations, it seems that our current model is not processing prior information in a very efficient way.

This was the first step of the model towards more complete understanding of variable interactions in laboratory experiments of atmospheric aging. More research and datasets are still needed to better quantify the dependence structure between studied variables. Recently, there has been various studies that have applied many dimension reduction techniques to oxidation chamber datasets showing that the selection of statistical dimension reduction technique can affect the interpretation of factors

received from dimension reduction of mass spectrometer dataset (Isokääntä et al., 2019; Koss et al., 2019; Rosati et al., 2019). For the purpose of emission oxidation study, a relevant way to form the groups would be to have the generation of oxidation products in the same factor. For the purpose of this study, the more explicit study of dimension reduction is out of the scope of this work. However, in the future development of the model, optimization of the choice of dimension reduction technique should be investigated more in detail.

**Code and data availability**

Codes are provided for reviewers and will be published in GitHub upon the publication of the manuscript. Data is partly uploaded to EUROCHAMP database ([https://data.eurochamp.org/data-access/chamber-experiments/](https://data.eurochamp.org/data-access/chamber-experiments/), chamber ILMARI / UEF, experiments Spruce log combustion aerosol + O3 - Aerosol study - physical properties) and will be completed before final publication of this manuscript.

**Author contribution**

SM, JK developed the idea of the study. PT, HC, OS, and AL carried out wood combustion experiments. VL, SM, JK, HC, PT and OS developed the model. VL and SI performed the data analysis. VL prepared the manuscript with contributions from all co-authors.

**Competing interests**

The authors declare that they have no conflict of interest.

**Acknowledgements**

Financial support. This work was supported by The Academy of Finland Centre of Excellence (grant no. 307331), The Academy of Finland Competitive funding to strengthen university research profiles (PROFI) for the University of Eastern Finland (grant no. 325022) and for University of Jyväskylä (grant no. 311877) and The Nessling foundation. Data collection

for this study has been partly funded from the European Union's 10 Horizon 2020 research and innovation programme through the EUROCHAMP-2020 Infrastructure Activity (grant No 730997)





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



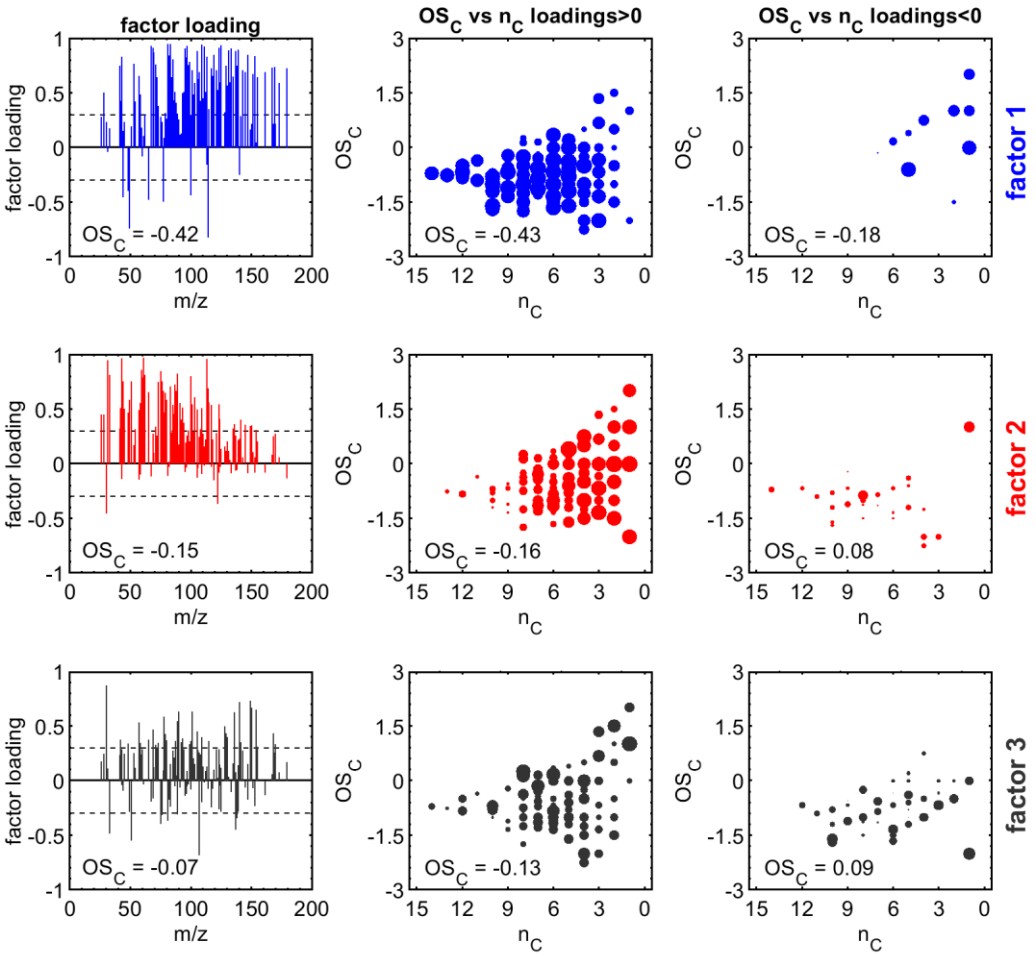

**Figure 1 Illustration of the three factor loadings (in rows) from exploratory factor analysis (EFA) including the average carbon oxidation state (OSC). Subplots in the left column contain the coefficients of the factor loadings and the limit of ±0.3 (dashed line), separating relevant from redundant variables. Subplots in the center (for m/z with positive factor loadings) and right column (for m/z with negative factor loadings) visualize the OS$_C$ dependent on the carbon number (n$_C$) of the detected sum formula (Kroll et al., 2011).**





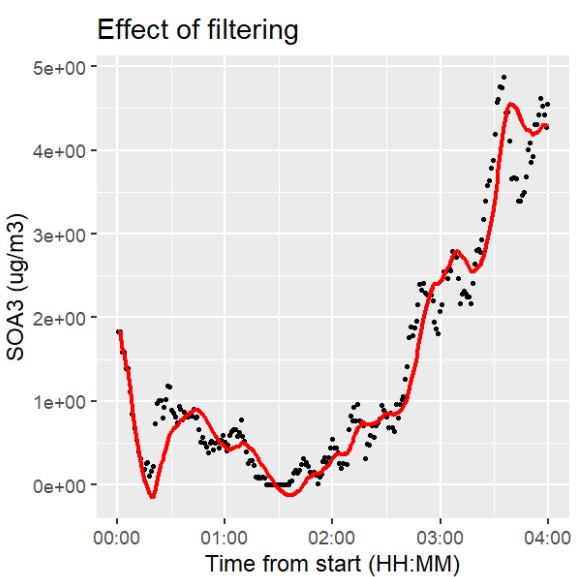

**Figure 2 Effect of filtering for one variable during dark aging period of experiment 2B. The dots represent the original measurements, and the line represents the filtered values.**

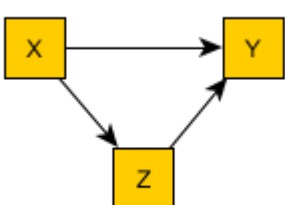

**Figure 3 Example of a causal graph. Edges between nodes (X, Y, and Z) describe the causal connection between variables and the direction of connections.**





(a)

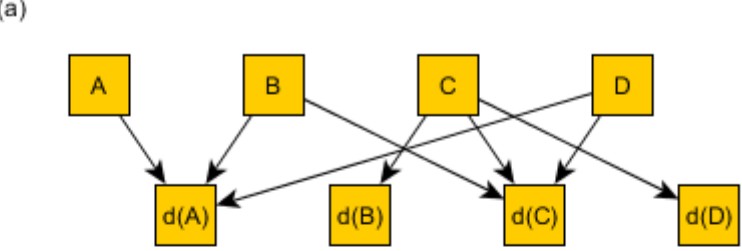

(b)

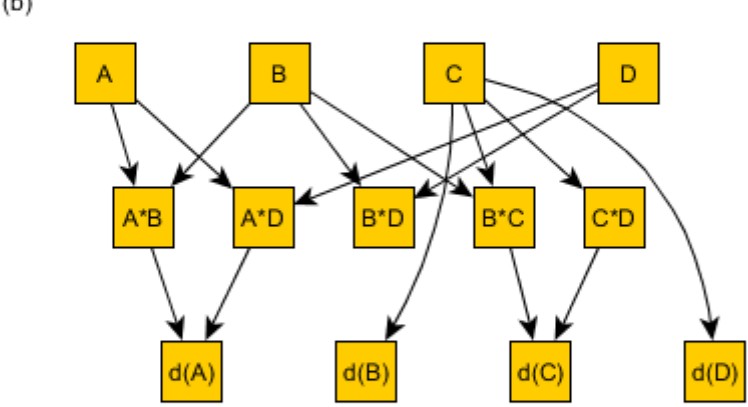

(c)

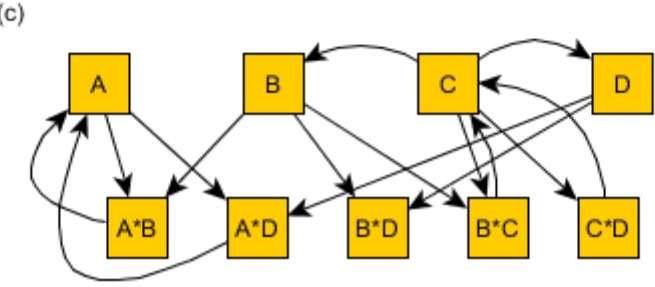

**Figure 4 (a) Variables are linked to other variables by affecting the $\Delta(x)$ (denoted as d(A) for variable A). Causal discovery algorithm searches for each $\Delta(x)$ the most probable variables that could have an effect. (b) How the interaction variables are linked to the edges are from causal discovery algorithm. All possible interaction variables are formed based on edges in graph a). For example, because A, B, and D are all affecting d(A), interaction variables A\*B, A\*D, and B\*D are formed, and those are possible to affect d(A). In this graph we assume that some of the interaction edges (e.g. B\*D -> d(A)) are not selected in LASSO. (c) Alternative way figure (b) can be drawn as a cyclic graph. Edge from e.g. A\*B -> d(A) in figure (b) is substituted by an edge A\*B -> A in figure (c). We have used this way to represent the edges in the results section.**



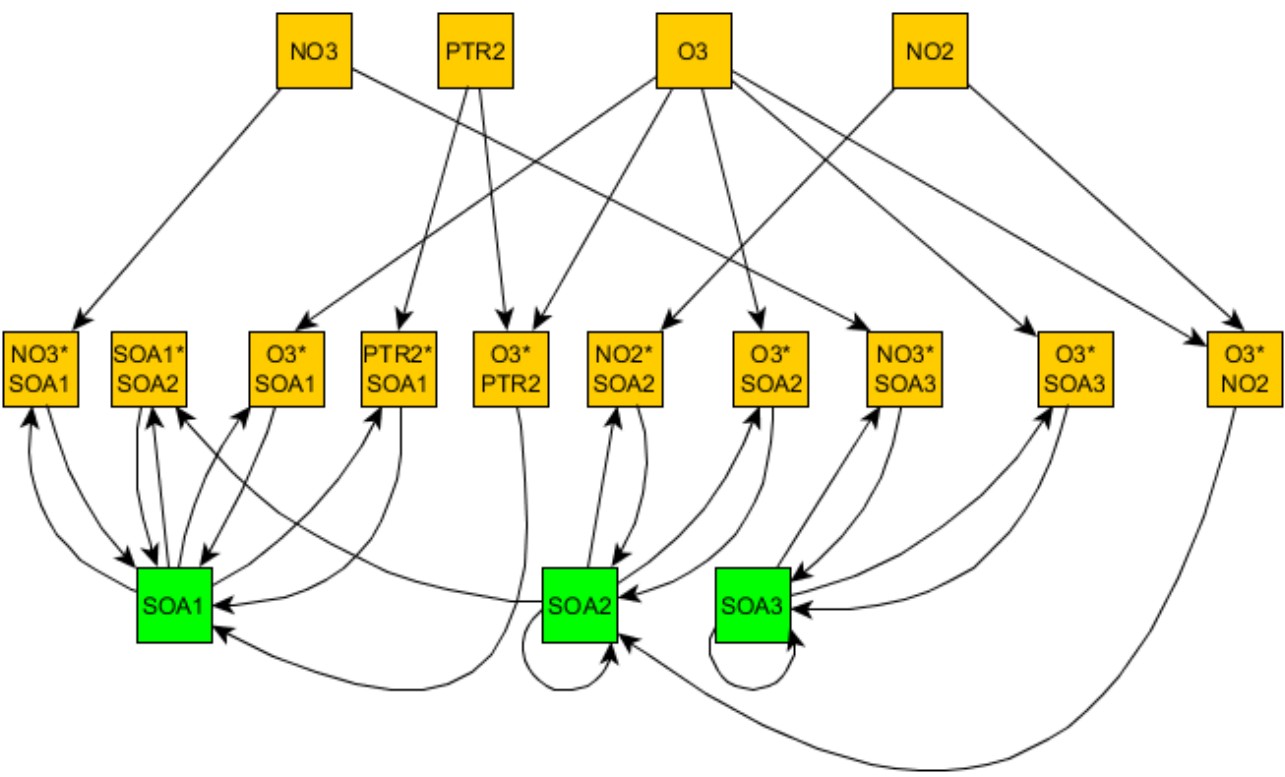

**Figure 5 SOA related part of the causal graph for dark aging experiments. SOA variables are highlighted in green.**

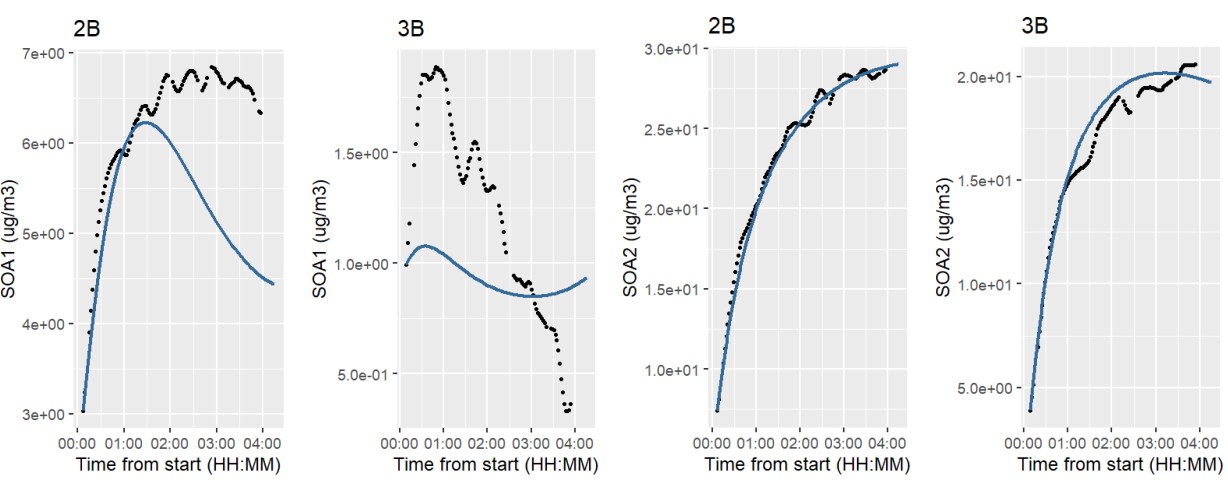

**Figure 6 Evolution of SOA factors 1 and 2 in dark aging experiments. Black points represent the filtered version of variable, and blue line is the modelled evolution.**



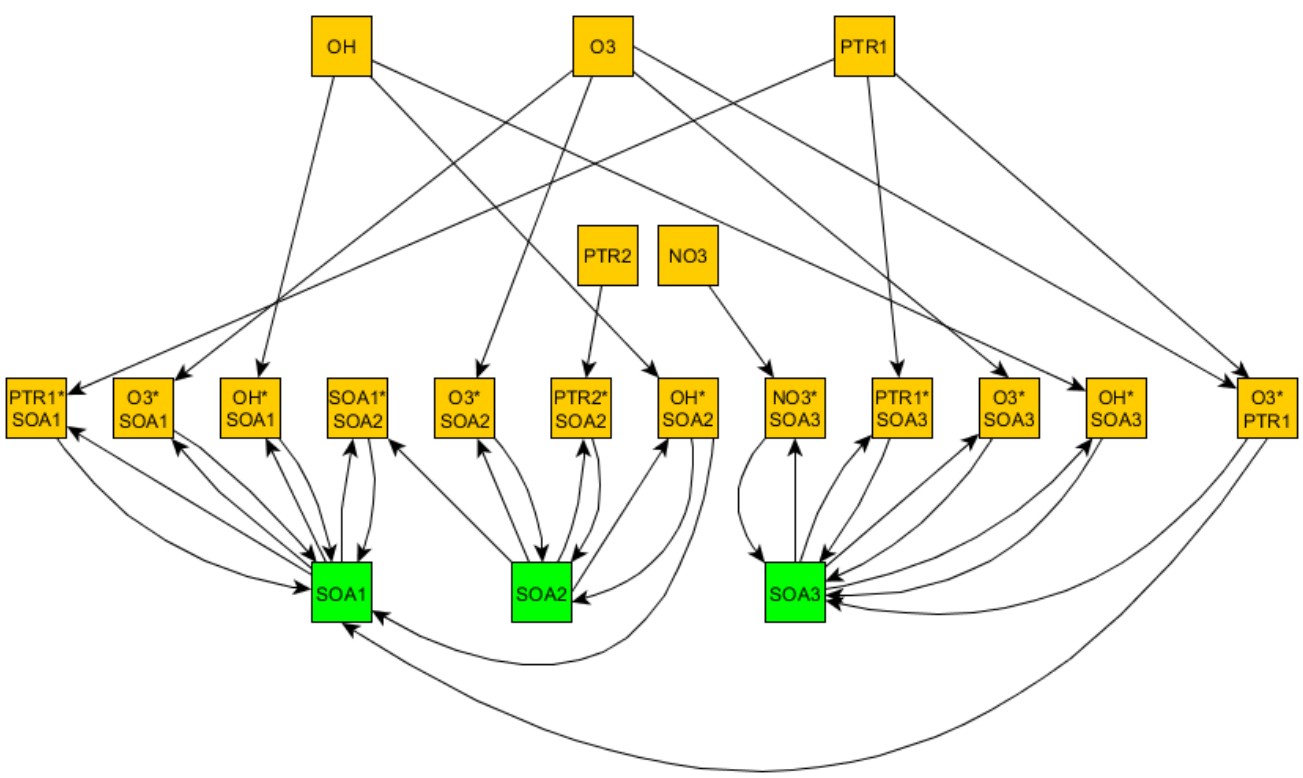

**Figure 7 SOA related part of the causal graph for photochemical aging experiments. SOA variables are highlighted in green.**

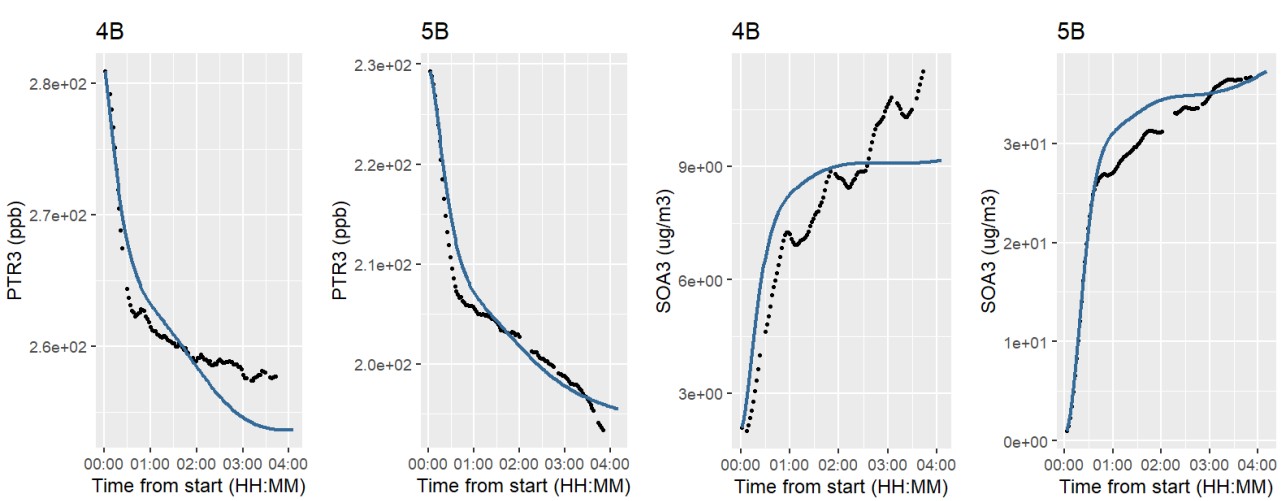

**Figure 8 Evolution of PTR3 and SOA3 factors in photochemical aging experiments. Black points represent the filtered version of variable, and blue line is the modelled evolution.**





**Table 1 Datasets used in this study. Wood combustion datasets are measured datasets from ILMARI environment (Tiitta et al., 2016). Simulated datasets are completely artificial datasets representing evolution of differential equation systems.**

|  | Dataset | How evolution is formed | Number of variables | Length of dataset (time points) |
|---|---|---|---|---|
| Wood combustion datasets | 2B and 3B | Dark aging experiments of wood combustion emission | 16 | ~100 |
|  | 4B and 5B | Photochemical aging experiments of wood combustion emission | 16 | ~100 |
| Simulated datasets | Small | Differential equation system using Mass Action Kinetics | 8 | 100+30 for prediction |
|  | Large | Differential equation system. Independent linear differential equations for each variable. System not follow Mass Action Kinetics | 16 | 100+30 for prediction |



**Table 2 Simulated experiments to study the model. How different options would affect model capability to return correct structure and goodness of fit.**

| Tests | How test was performed | Purpose of the test |
|---|---|---|
| Measurement frequency (Table 3 and 4) | Reducing or increasing number of simulated 'measurement' points in given time (100). Frequency of 1/4,1/2,1,2,4 time points was used. This means dataset has 401, 201, 101, 51 and 26 measurement points during time simulation time 0-100. | To see whether increasing or decreasing measurement frequency would help making model better. Increasing of points with same measurement noise would lead to lower signal/noise ratio? |
| Measurement uncertainty (Table 3 and 4) | Adding normally distributed random noise (sd = unc_frac) to the simulated evolution. This random noise was representing the possible measurement uncertainty. | Purpose was to see if the measurement uncertainty reduces both structural accuracy and fit accuracy. |
| Filtering and smoothing (Table 5) | Applied filtering and smoothing to original variables of simulated dataset. | How filtering or smoothing would improve the fit and structure of the model? Is it reasonable to use filtering or smoothing of dataset before making a model? |
| Prior Information (Table 6 and 7) | Using prior information about the edges possible in the model. Using both correct and incorrect prior information. Fraction of information from all correct prior information was used as a measure of information. | Does addition of prior information help model to get correct structure and good fit? |





**Table 3 Effect of measurement frequency and fraction of error to goodness of fit –parameters (Fscore, RMSE) of the model for larger simulated dataset. Variable nEdges describes the number of edges in each model (single and interaction variables affecting $\Delta(x)$s). Mean average error of the predicted evolution of the model was calculated (RMSE_pred). CorMean is the mean correlation between correct edge and the most correlated edge in the model. For each row, number of replications is 10.**

| nObs | unc_fraction | nEdges.mean | RMSE.mean | RMSE_pred.mean | Fscore.mean | corMean.mean |
|------|------|------|------|------|------|------|
| 26 | 0,01 | 35,9 | 17,07 | 12,86 | 0,22 | 0,88 |
| 51 | 0,01 | 46,2 | 17,47 | 16,2 | 0,22 | 0,88 |
| 101 | 0,01 | 41,1 | 19,43 | 18,13 | 0,22 | 0,87 |
| 201 | 0,01 | 39,1 | 23 | 20,63 | 0,2 | 0,86 |
| 401 | 0,01 | 28 | 24,46 | 17,85 | 0,21 | 0,83 |
| 26 | 0,05 | 30 | 19,97 | 16,51 | 0,19 | 0,82 |
| 51 | 0,05 | 29,9 | 19,77 | 20,07 | 0,17 | 0,84 |
| 101 | 0,05 | 29,5 | 17,05 | 19,03 | 0,19 | 0,89 |
| 201 | 0,05 | 27,4 | 19,84 | 22,21 | 0,19 | 0,92 |
| 401 | 0,05 | 24,5 | 23,61 | 20,71 | 0,18 | 0,93 |
| 26 | 0,1 | 25,9 | 19,96 | 15,68 | 0,17 | 0,86 |
| 51 | 0,1 | 28,9 | 18,91 | 19,51 | 0,2 | 0,93 |
| 101 | 0,1 | 28 | 21,24 | 21,56 | 0,19 | 0,93 |
| 201 | 0,1 | 22,4 | 28,06 | 35,83 | 0,18 | 0,92 |
| 401 | 0,1 | 19,8 | 33,86 | 22,49 | 0,18 | 0,91 |
| 26 | 0,5 | 33,7 | 31,17 | 25,76 | 0,19 | 0,93 |
| 51 | 0,5 | 28,3 | 47,12 | 31,35 | 0,19 | 0,92 |
| 101 | 0,5 | 41,1 | 1757,27 | 38,03 | 0,22 | 0,94 |
| 201 | 0,5 | 44,3 | 344,31 | 42,27 | 0,23 | 0,94 |
| 401 | 0,5 | 46,1 | 2,19841E+18 | 71,84 | 0,23 | 0,93 |



**Table 4 Effect of measurement frequency and fraction of error to fitting parameters in smaller simulated dataset for the goodness of fit –parameters (Fscore, RMSE) of the model for smaller simulated dataset. Variable nEdges describes the number of edges in each model (single and interaction variables affecting $\Delta(x)$s). Mean average error of the predicted evolution of the model was calculated (RMSE_pred). CorMean is the mean correlation between correct edge and the most correlated edge in the model. For each row, number of replications is 10.**

| nObs | unc_fraction | nEdges.mean | RMSE.mean | RMSE_pred.mean | Fscore.mean | corMean.mean |
|---|---|---|---|---|---|---|
| 26 | 0,01 | 15,3 | 7,76 | 4,5 | 0,22 | 0,85 |
| 51 | 0,01 | 17,7 | 5,88 | 4,16 | 0,23 | 0,87 |
| 101 | 0,01 | 22,5 | 4,64 | 3,67 | 0,21 | 0,81 |
| 201 | 0,01 | 22,5 | 4,02 | 3,71 | 0,21 | 0,75 |
| 401 | 0,01 | 18,4 | 6,68 | 4,07 | 0,26 | 0,81 |
| 26 | 0,05 | 12,6 | 8,79 | 4,71 | 0,13 | 0,75 |
| 51 | 0,05 | 14,1 | 8,93 | 3,89 | 0,2 | 0,79 |
| 101 | 0,05 | 16,8 | 7,89 | 3,33 | 0,25 | 0,81 |
| 201 | 0,05 | 13,8 | 10,83 | 2,85 | 0,19 | 0,8 |
| 401 | 0,05 | 11,5 | 17,93 | 3,9 | 0,24 | 0,8 |
| 26 | 0,1 | 10,1 | 9,91 | 3,59 | 0,17 | 0,77 |
| 51 | 0,1 | 10,6 | 10,99 | 2,92 | 0,24 | 0,78 |
| 101 | 0,1 | 11,3 | 15,23 | 4,35 | 0,24 | 0,79 |
| 201 | 0,1 | 9,8 | 22,61 | 8,1 | 0,31 | 0,68 |
| 401 | 0,1 | 7,4 | 32,08 | 11,81 | 0,32 | 0,63 |
| 26 | 0,5 | 5,6 | 31,64 | 12,85 | 0,23 | 0,61 |
| 51 | 0,5 | 5 | 40,16 | 21,21 | 0,23 | 0,59 |
| 101 | 0,5 | 6 | 47,46 | 29,68 | 0,22 | 0,63 |
| 201 | 0,5 | 6,4 | 51,29 | 36,34 | 0,23 | 0,65 |
| 401 | 0,5 | 8,2 | 2,5E+17 | 795977,5 | 0,26 | 0,76 |




**Table 5 Effect of filtering and smoothing techniques to the goodness of fit –parameters (Fscore, RMSE) of the model for larger simulated dataset. Filter1 is a version, where filtering window was chosen manually for each time series, filter2 has automatic selection of filtering window, and smooth when smoothing was used with automatic selection of filtering window. Row 1-7 are results for raw dataset where any technique haven't been applied. Variable nEdges describes the number of edges in each model (single and interaction variables affecting Δ(x)s). Mean average error of the predicted evolution of the model was calculated (RMSE_pred). CorMean is the mean correlation between correct edge and the most correlated edge in the model. For each row, number of replications is 10.**

| unc_fraction | smooth | filter1 | filter2 | nEdges.mean | RMSE.mean | RMSE_pred.mean | Fscore.mean | corMean.mean |
|---|---|---|---|---|---|---|---|---|
| 0,01 | 0 | 0 | 0 | 50,15 | 17,24 | 27,79 | 0,22 | 0,92 |
| 0,05 | 0 | 0 | 0 | 40,4 | 16,94 | 22,09 | 0,2 | 0,92 |
| 0,075 | 0 | 0 | 0 | 40,45 | 17,93 | 24,35 | 0,21 | 0,93 |
| 0,1 | 0 | 0 | 0 | 42,15 | 18,74 | 28,54 | 0,22 | 0,94 |
| 0,3 | 0 | 0 | 0 | 42,9 | 25,31 | 31,29 | 0,22 | 0,94 |
| 0,5 | 0 | 0 | 0 | 39,2 | 56,24 | 155,92 | 0,21 | 0,92 |
| 1 | 0 | 0 | 0 | 44,85 | 132238,3 | 62,34 | 0,2 | 0,94 |
| 0,01 | 1 | 0 | 0 | 53,7 | 16,35 | 25,33 | 0,22 | 0,95 |
| 0,05 | 1 | 0 | 0 | 52,5 | 15,84 | 23,43 | 0,23 | 0,95 |
| 0,075 | 1 | 0 | 0 | 53,6 | 15,54 | 21,67 | 0,24 | 0,94 |
| 0,1 | 1 | 0 | 0 | 55,9 | 15,69 | 9288,04 | 0,26 | 0,95 |
| 0,3 | 1 | 0 | 0 | 48,5 | 18,22 | 23,81 | 0,23 | 0,94 |
| 0,5 | 1 | 0 | 0 | 44,3 | 19,33 | 157,1 | 0,23 | 0,92 |
| 1 | 1 | 0 | 0 | 47,2 | 4,55E+60 | 142,6 | 0,21 | 0,94 |
| 0,01 | 0 | 1 | 0 | 45,8 | 16,49 | 21,88 | 0,24 | 0,91 |
| 0,05 | 0 | 1 | 0 | 33,8 | 21,75 | 25,93 | 0,19 | 0,84 |
| 0,075 | 0 | 1 | 0 | 38,8 | 20,26 | 79,84 | 0,2 | 0,89 |
| 0,1 | 0 | 1 | 0 | 38,3 | 19,81 | 29,76 | 0,2 | 0,9 |
| 0,3 | 0 | 1 | 0 | 30,7 | 22,8 | 318828,5 | 0,18 | 0,88 |
| 0,5 | 0 | 1 | 0 | 39 | 29,32 | 113,67 | 0,2 | 0,92 |
| 1 | 0 | 1 | 0 | 62,3 | 7,95E+11 | 7103,91 | 0,25 | 0,96 |
| 0,01 | 0 | 0 | 1 | 48,4 | 15,96 | 24,38 | 0,23 | 0,93 |
| 0,05 | 0 | 0 | 1 | 38 | 19,58 | 41,03 | 0,21 | 0,89 |
| 0,075 | 0 | 0 | 1 | 42,1 | 19,57 | 28,03 | 0,21 | 0,9 |
| 0,1 | 0 | 0 | 1 | 41,2 | 20,47 | 34,79 | 0,21 | 0,89 |
| 0,3 | 0 | 0 | 1 | 33,5 | 22,31 | 194,23 | 0,2 | 0,9 |
| 0,5 | 0 | 0 | 1 | 46,3 | 25,96 | 106945,1 | 0,22 | 0,94 |
| 1 | 0 | 0 | 1 | 59,6 | 64,11 | 56,9 | 0,24 | 0,96 |



**Table 6 Effect of correct prior information in smaller simulated dataset. Fraction of correct prior information about correct (FracCC) and incorrect (FracCI) edges in smaller simulated dataset for goodness of fit –parameters (Fscore, RMSE) of the model. Variable nEdges describes the number of edges in each model (single and interaction variables affecting $\Delta(x)$s).Mean average error of the predicted evolution of the model was calculated (RMSE_pred). CorMean is the mean correlation between correct edge and the most correlated edge in the model. For each row, number of replications is 10.**

| fracCC | fracCI | fracIC | fracII | nEdges.mean | RMSE.mean | RMSE_pred.mean | Fscore.mean | corMean.mean |
|---|---|---|---|---|---|---|---|---|
| 0,05 | 0,05 | 0 | 0 | 10,4 | 15,85 | 4,98 | 0,2 | 0,79 |
| 0,3 | 0,05 | 0 | 0 | 22,1 | 8,99 | 4,85 | 0,36 | 0,87 |
| 0,5 | 0,05 | 0 | 0 | 31,2 | 7,62 | 4,74 | 0,44 | 0,9 |
| 0,05 | 0,3 | 0 | 0 | 11 | 15,36 | 4,95 | 0,23 | 0,8 |
| 0,3 | 0,3 | 0 | 0 | 20,7 | 12,31 | 4,83 | 0,36 | 0,83 |
| 0,5 | 0,3 | 0 | 0 | 30,9 | 7,07 | 4,54 | 0,43 | 0,9 |
| 0,05 | 0,5 | 0 | 0 | 10,8 | 14,55 | 4,93 | 0,23 | 0,8 |
| 0,3 | 0,5 | 0 | 0 | 22 | 6,73 | 5,2 | 0,38 | 0,88 |
| 0,5 | 0,5 | 0 | 0 | 32,3 | 5,19 | 5,33 | 0,48 | 0,93 |





**Table 7 Effect of incorrect prior information in smaller simulated dataset. Fraction of incorrect prior information about correct (FracIC) and incorrect (FracII) edges in smaller simulated dataset for goodness-of-fit –parameters (Fscore, RMSE) of the model. Variable nEdges describes the number of edges in each model (single and interaction variables affecting $\Delta(x)$s).Mean average error of the predicted evolution of the model was calculated (RMSE_pred). CorMean is the mean correlation between correct edge and the most correlated edge in the model. For each row, number of replications is 10.**

| fracCC | fracCI | fracIC | fracII | nEdges.mean | RMSE.mean | RMSE_pred.mean | Fscore.mean | corMean.mean |
|---|---|---|---|---|---|---|---|---|
| 0,3 | 0,3 | 0 | 0 | 20,7 | 12,31 | 4,83 | 0,36 | 0,83 |
| 0,3 | 0,3 | 0,3 | 0 | 32,5 | 6,2 | 5,07 | 0,45 | 0,9 |
| 0,3 | 0,3 | 0,5 | 0 | 41,9 | 2,23 | 5,15 | 0,51 | 0,95 |
| 0,3 | 0,3 | 0 | 0,3 | 21,1 | 9,14 | 4,59 | 0,38 | 0,87 |
| 0,3 | 0,3 | 0,3 | 0,3 | 33 | 5,51 | 5,61 | 0,49 | 0,91 |
| 0,3 | 0,3 | 0,5 | 0,3 | 44,6 | 3,16 | 5,96 | 0,5 | 0,95 |
| 0,3 | 0,3 | 0 | 0,5 | 22 | 10,5 | 4,62 | 0,4 | 0,86 |
| 0,3 | 0,3 | 0,3 | 0,5 | 32,7 | 5,34 | 5,09 | 0,48 | 0,92 |
| 0,3 | 0,3 | 0,5 | 0,5 | 44,2 | 2,43 | 4,6 | 0,51 | 0,95 |