# Peer review of "Atmospheric aging of small-scale wood combustion emissions (model MECHA 1.0) – is it possible to distinguish causal effects from non-causal associations?"

_Geoscientific Model Development, 2020_

## Referee Comment (RC1) · Anonymous Referee #1 · 14 Mar 2020

Leinonen et al. have developed a first-of-its-kind semi-empirical model to simulate the gas- and particle-phase evolution of organic species while aging emissions from wood combustion. They comment on model performance and discuss the advantages and disadvantages of using such a model to assess causality between the predictors and outcomes.

There are several key deficiencies in this manuscript and I do not recommend publication of this manuscript to GMD. The primary deficiency of this manuscript is that the methods and application for the MECHA model are very poorly described and are very

hard to follow (assuming I represent the average researcher within the atmospheric modeling community). I suspect if the methods need to be described and published in a more appropriate journal and that this manuscript should focus primarily on the application to aging of wood combustion emissions. For instance, sections 2.2 to 2.5.4 were quite abstract. A lot of technical terms were introduced (e.g., causal discovery algorithm, causal graph) but there wasn't a detailed explanation of how its use applies to the application explored in this work. Another deficiency is that the technical communication is grammatically incorrect in a lot of places, which it makes it hard to follow the authors' train of thought. Overall, it was very hard to assess the scientific merit of this work.

It is also quite likely that I am not well versed with this type of modeling (to me, the model appears to be one where one solves a set of differential equations on aggregated variables, which are factors from the gas and particle instrument data, to fit coefficients that can reproduce the observed data; a related question then is how was the model was trained and tested?). If that is true, I might not be well-suited to review this manuscript.

Regardless, I feel confident that the average reader of this journal is going to be hard pressed to understand the model and the key findings from this work.
* * *

---

## Short Comment (SC1) · 16 May 2020

This is an executive editor comment highlighting the ways in which this manuscript is not currently compliant with GMD policy on code and data availability.

In the case of this manuscript, the issues are exceptionally serious and need to be addressed immediately. To quote from the GMD model code and data policy: [1]
* * *
[1] https://www.geoscientific-model-development.net/about/code_and_data_policy.html

> Code and data access must be provided at the time that the discussion paper is submitted. Embargoes, whether pending acceptance or for a defined period, are not acceptable.

Here, the authors have completely ignored this rule by failing to provide any code at all, and only providing incomplete data. This undermines the open review process of GMD and needs to be remedied forthwith. Specifically, the full code and data must be publicly and persistently archived and cited from the code and data availability section. Given that these citations have been omitted from the manuscript, the authors need to rapidly post an author comment citing the code and data. If this is not possible the manuscript is not yet ready for peer review and should be rejected.

As an aside, I should point out that the code and data availability policy specifically states that GitHub is not a suitable archival site. Indeed, even GitHub tell users to use Zenodo for this purpose[2].

The authors should familiarise themselves with the code and data policy and need to ensure that the manuscript complies with it fully. Further explanation of the policy is contained in the connected editorial: https://doi.org/10.5194/gmd-12-2215-2019.
* * *
[Figure]
* * *
[2]https://guides.github.com/activities/citable-code/

---

## Referee Comment (RC2) · Anonymous Referee #2 · 18 May 2020

This paper attempts to describe a semi-empirical model to simulate the evolution of gas-phase and particle phase measurements of aging emissions from wood combustion in the ILMARI atmospheric simulation chamber. The authors attempt to describe the performance of their model by employing different techniques (e.g. adding noise to their data, smoothing in different ways, filtering the data in different ways).

I find several troublesome aspects of the manuscript. The language/grammar is not clear and is often confusing. It made it difficult to evaluate portions of the manuscript. Further, details regarding the modelling aspects were often introduced with little to no

explanation (or background) and in ambiguous terms. Rather than concrete examples of how the model was treating the data. The tables in the text were not clear nor adequately explained making it difficult to assess what data was included and the parameters varied. Honestly, it appeared as if the modelling output was directly placed into a table with no thought to whether or not the information presented was considered useful or not.

Overall, I find the text challenging to disentangle and is not suitable to be published in the current form. It requires significant grammatical polishing and a more extensive discussion regarding the details of the model so that it could be useful for others. I also find it difficult to assess its usefulness to the chamber community writ large.

Minor comments / questions can be found in the supplement.

Please also note the supplement to this comment:
https://www.geosci-model-dev-discuss.net/gmd-2020-13/gmd-2020-13-RC2-supplement.pdf

**Supplement:**

Other comments:

Line 14 page 2: "to double to triple" this is a bit of a colloquialism.

Line 16 page 2: PAH**s**

Line 29 page 2: … are important and are needed to …

Line 1 page 3: if a few attempts have been made it might be nice to cite them here.

Line 3 page 3: the V in Volatility doesn't need to be capitalized

Lines 3 – 6 page 3: this sentence needs considerable work.

Perhaps: In the VBS approach, the evolution of the constituent phases (gas and particle) are modelled based on the volatilities of the compounds, …

Lines 6 – 11 page 3: I think there needs to be a link between SOA and gas phase oxidation when discussing MCM. An approach to model SOA is to utilize MCM for the production of oxidized molecules with a parameterization based molecular formula / function group analysis to estimate saturation vapor concentrations, etc… This is what you do when talking about the SOM, but MCM itself is not a way to model SOA.

Line 14 page 3: "All approaches …. differential equation approaches." Consider rewording

Line 16 page 3: remove: "here the amount of SOA" or reword. The emphasis on here the amount of SOA is not clear the context. Is this the point of the current paper or the current topic you are introducing (SOA models)?

Line 18 page 3: "In this kind of system" Please describe what you mean here.

Line 21 page 3: "Initial compounds cause the increase in products and decrease in their own concentrations" maybe add at the beginning "If a reaction is favorable, the initial compounds result in an increase in the products and a decrease in their own concentrations."

Page 4: it is mentioned that O3 is put into the chamber to simulate dark aging, where dark aging represents both O3 and NO3 chemistry. It is not clear what concentrations of NOx are in the chamber and why NO3 chemistry would be taking place. It could be helpful to have an experimental table listing the types of experiments taking place and concentrations of relevant oxidants / trace gases.

Line 26 page 4: "intensive" should be 'the intense'

Line 28 page 4: "their formation products" the formation of what products? HNO3? Molecules making up SOA?

Line 1 page 5: can you clarify how it impacts the composition? Just one line.

Lines18 – 24 page 5: how are the factors determined?

Line 8 page 6: " but EFA was selected **for** further analysis"

Line 2 page 7: "OH radical has an …" should be: "OH radicals have an …"

Section 2.1.2 within Barmet et al. the OH concentration needs to be determine with reference to another VOC with a known rate constant. For instance:

$$OH\ Exposure = \left( \frac{ln\left(\frac{d_9 butanol}{naphthalene}\right)_0 - ln\left(\frac{d_9 butanol}{naphthalene}\right)_t}{k_{OH,butanol} - k_{OH,naphthalene}} \right)$$

What was the other VOC chosen for comparison?

Line 29 page 9: "evolvement" should be evolution

Lines 30 – 31 page 10: is it reasonable that OH chemistry is not allowed to occur during dark aging? If there is ozonolysis of alkenes taking place, then there will be OH radicals produced. Even if it is a small pathway it shouldn't be ruled out unless there was negligible dark aging in the presence of O3 alone (without $NO_3$ chemistry taking place), or if you could model NO3 production to show there is negligible ozonolysis occurring.

Line 15 Page 11: I believe it should say "physical **properties**" because I don't understand how physical reactions would be what is meant.

Line 32 Page 11 / Line 1 Page 12: should read " which assessed the measurement error…" omit was.

Line 8 page 12: replace "much" with many

Line 8 page 12: omit the sentence starting with "Additionally" either talk about how it was interesting or remove the sentence.

Line 10/11 page 12: what is "correct structure"?

Line 18 page 12: "which" should be replaced with 'whose'

Line 20 page 12: what are Mass Action Kinetics system?

Line 31 page 12: RMSE is not defined. (I presume root mean square error, and will assume that is what is meant)

Lines 33 – 34 page 12: Why is this weighting required? Why would some time series not have a standard deviation (standard deviation of what?)? This is not explained adequately.

Lines 3-4 page 13: Why is only 30% of the simulated dataset predicted?

Section 3.1

Lines 24 – 31 page 14: The problem I have with this paragraph is its order. 1) you show the measurement of error in the Tables. 2) there is good agreement! 3) This isn't surprising. 4) discuss the actual data.

I would suggest rearranging the content so you first say this table shows what you want to show. Then discuss those results, then talk about the goodness of agreement and the overall conclusion.

Tables 3-5 and not intuitive and are not adequately described. For instance, what is "unc_fraction"? or what is nObs? This information is not included in the Table headings. Maybe the tables should be

adapted so they don't appear as if they were just directly taken from a code output. For example: corMean.mean does not convey helpful information. Purely call the column title "Correlation Mean"

Line 31 page 15, what is fittingness?

Figure 5: are there not background measurements? The Figures should all start at 0 on the y-axis. Why do the figures not start at 0 mass loading? What is going on before the initiating of dark aging, are the concentrations stable? Why is the agreement so poor for 3B SOA 1?

Lines 13-14 page 16, why would $NO_3$ radicals change SOA1 to SOA2? If nitrate is important in this process, then it would be condensed phase process where $N2O5$ could be uptaken into the particle phase.

Line 14 page 16: why is NO2 attributed here and not NO3 radicals? Because in the gas-phase that is when NO3 chemistry would be important.

---

## Author Comment (AC1) · 20 May 2020

We thank the executive editor for the comment. As all of the data were not ready for publication in Eurochamp database where it will be stored, we promised the handling topical editor to send the data and codes to him to share with the reviewers, if they would require them. We misunderstood the core principle 2 of guidelines.

The codes and data will now be available in Zenodo, the link will be provided in separate short comment.

---

## Author Comment (AC2) · 20 May 2020

All data and codes are now available in Zenodo. Link:
http://doi.org/10.5281/zenodo.3835482
* * *

---

## Author Comment (AC3) · 7 Aug 2020

We thank the reviewer for the review and constructive comments. Below we address each comment point by point. Reviewer comments are marked as black and our response as blue.

Leinonen et al. have developed a first-of-its-kind semi-empirical model to simulate the gas- and particle-phase evolution of organic species while aging emissions from wood combustion. They comment on model performance and discuss the advantages and disadvantages of using such a model to assess causality between the predictors and outcomes.

There are several key deficiencies in this manuscript and I do not recommend publication of this manuscript to GMD. The primary deficiency of this manuscript is that the methods and application for the MECHA model are very poorly described and are hard to follow (assuming I represent the average researcher within the atmospheric modeling community). I suspect if the methods need to be described and published in a more appropriate journal and that this manuscript should focus primarily on the application to aging of wood combustion emissions. For instance, sections 2.2 to 2.5.4 were quite abstract. A lot of technical terms were introduced (e.g., causal discovery algorithm, causal graph) but there wasn't a detailed explanation of how its use applies to the application explored in this work. Another deficiency is that the technical communication is grammatically incorrect in a lot of places, which it makes it hard to follow the authors' train of thought. Overall, it was very hard to assess the scientific merit of this work.

It is also quite likely that I am not well versed with this type of modeling (to me, the model appears to be one where one solves a set of differential equations on aggregated variables, which are factors from the gas and particle instrument data, to fit coefficients that can reproduce the observed data; a related question then is how was the model was trained and tested?). If that is true, I might not be well-suited to review this manuscript.

Regardless, I feel confident that the average reader of this journal is going to be hard pressed to understand the model and the key findings from this work.

The reviewer is correct, the model is based on solving multiple differential equations on aggregated variables. To clarify the methods section, we decided to remove chapters 2.2-2.4 and focused more on connecting causal discovery algorithms and differential equations. Due these changes, the terminology is now somewhat simplified and thus it should be easier to understand for readers applying these types of models.

Model performance was tested on simulated datasets, where the connections between variables were known. Because we had only few chamber

experiments, we decided to use those entirely to form the structure. Obtained structures for dark and UV ageing experiments could be tested in future studies.

---

## Author Comment (AC4) · 7 Aug 2020

We thank the reviewer for the review and constructive comments. Below we address each comment point by point. Reviewer comments are marked as black, our response as blue and corrections to the changes to the manuscript as red.

This paper attempts to describe a semi-empirical model to simulate the evolution of gas-phase and particle phase measurements of aging emissions from wood combustion in the ILMARI atmospheric simulation chamber. The authors attempt to describe the performance of their model by employing different techniques (e.g. adding noise to their data, smoothing in different ways, filtering the data in different ways).

I find several troublesome aspects of the manuscript. The language/grammar is not clear and is often confusing. It made it difficult to evaluate portions of the manuscript. Further, details regarding the modelling aspects were often introduced with little to no explanation (or background) and in ambiguous terms. Rather than concrete examples of how the model was treating the data. The tables in the text were not clear nor adequately explained making it difficult to assess what data was included and the parameters varied. Honestly, it appeared as if the modelling output was directly placed into a table with no thought to whether or not the information presented was considered useful or not.

Overall, I find the text challenging to disentangle and is not suitable to be published in the current form. It requires significant grammatical polishing and a more extensive discussion regarding the details of the model so that it could be useful for others. I also find it difficult to assess its usefulness to the chamber community writ large. Minor comments / questions can be found in the supplement.

To clarify the methods section, we decided to remove chapters 2.2-2.4 and focused more on connecting causal discovery algorithms and differential equations. Due these changes, the terminology is now somewhat simplified and thus it should be easier to understand for readers applying these types of models.

All tables are necessary for model evaluation and all variables presented in tables are introduced in section 2.3 in the revised manuscript. We have clarified the tables form to be easier to interpret (see detailed comment below).

Other comments:

Line 14 page 2: "to double to triple" this is a bit of a colloquialism.
Changed to "...enhance the organic aerosol concentration between a factor of two to three..."

Line 16 page 2: PAHs
Corrected

Line 29 page 2: ... are important and are needed to ...
Corrected

Line 1 page 3: if a few attempts have been made it might be nice to cite them here.

We added some references to this (Hartikainen et al., 2018, 2020; Isaacman-Vanwertz et al., 2018)

Line 3 page 3: the V in Volatility doesn't need to be capitalized
Corrected

Lines 3 – 6 page 3: this sentence needs considerable work.

Perhaps: In the VBS approach, the evolution of the constituent phases (gas and particle) are modelled based on the volatilities of the compounds, ...
Corrected as suggested

Lines 6 – 11 page 3: I think there needs to be a link between SOA and gas phase oxidation when discussing MCM. An approach to model SOA is to utilize MCM for the production of oxidized molecules with a parameterization based molecular formula / function group analysis to estimate saturation vapor concentrations, etc... This is what you do when talking about the SOM, but MCM itself is not a way to model SOA.
This was now corrected as
"Another approach to model SOA and especially its precursors in the gas-phase is the family of explicit chemical modeling. There exists several chemical models such as Master Chemical Mechanism (MCM) (Jenkin et al., 1997; Saunders et al., 2003) and GECKO-A (Aumont et al., 2005), which comprise large amounts of chemical reactions and pre-determined reaction coefficients to replicate the evolution of the system. MCM has been recently applied to wood burning emissions by running the model with most important primary emission species to model the evolution of gas-phase species using

smaller selection of reactions from the whole system (Coggon et al., 2019). These can be used to parametrize SOA production. …"

Line 14 page 3: "All approaches .... differential equation approaches."
Consider rewording
Reworded as "All approaches, volatility-based, SOM, and explicit chemical modeling, are based on differential equations."

Line 16 page 3: remove: "here the amount of SOA" or reword. The emphasis on here the amount of SOA is not clear the context. Is this the point of the current paper or the current topic you are introducing (SOA models)?
Removed

Line 18 page 3: "In this kind of system" Please describe what you mean here.
System was meant to be one of the approaches mentioned above. This is now clarified as "In the approaches mentioned above …"

Line 21 page 3: "Initial compounds cause the increase in products and decrease in their own concentrations" maybe add at the beginning "If a reaction is favorable, the initial compounds result in an increase in the products and a decrease in their own concentrations."
Corrected as suggested

Page 4: it is mentioned that O3 is put into the chamber to simulate dark aging, where dark aging represents both O3 and NO3 chemistry. It is not clear what concentrations of NOx are in the chamber and why NO3 chemistry would be taking place. It could be helpful to have an experimental table listing the types of experiments taking place and concentrations of relevant oxidants / trace gases.

Although NO3 radicals had not been measured directly, the high concentrations of NO2, which was previously converted from NO by adding ozone, and excess ozone should lead to NO3 radical formation. In addition, during dark aging (absence of OH-radicals) an extensive decay of phenolic and furanoic VOCs was observed. Since the reaction rates of phenolic and furanoic compounds with ozone are low, it can be concluded that NO3 radicals were present (Hartikainen et al., 2018). Further, extensive formation of nitrophenols was observed as a result of reactions between NO3 and phenolic precursors (Hartikainen et al., 2018).

We added the following text to the revised manuscript line 19-22 page 4:

"The conditions in the chamber simulate polluted atmospheric boundary-layer conditions with an OH concentration of (0.5–5) x $10^6$ molec cm$^{-3}$, ozone concentrations of 20–90 ppb and NOx concentrations of 40–120 ppb (Tiitta et al., 2016) with a lower VOC-to-NOx ratio (fast ignition: ratio $\approx$ 3) yielded smaller total emissions including SOA than the slow ignition cases (ratio $\approx$ 5)."

Line 26 page 4: "intensive" should be 'the intense'
Corrected

Line 28 page 4: "their formation products" the formation of what products? HNO3? Molecules making up SOA?

Differences in dark and light ageing has been mentioned in lines 12-23 page 4 in revised manuscript. Text has been also clarified by removing one sentence.

Line 1 page 5: can you clarify how it impacts the composition? Just one line.

The first sentence is reworded and second sentence was added as
"The ignition type, i.e. how fast the logwood ignites change influence the emission factors of POA and VOC emission from wood combustion, in particular carbonyls, aromatic hydrocarbons, furanoic and phenolic compounds (Hartikainen et al., 2018; Tiitta et al., 2016). During aging furanoic and phenolic compounds decreased and nitrogen-containing organic compounds in both gas and particulate phase were produced. Photochemical aging increased especially the concentrations of certain gaseous carbonyls, particularly acid anhydrides (Hartikainen et al., 2018)."

Lines18 – 24 page 5: how are the factors determined?

More detailed description on the factor determination will be given in revised manuscript lines 15-33, page 6.

Line 8 page 6: " but EFA was selected for further analysis"
Corrected

Line 2 page 7: "OH radical has an ..." should be: "OH radicals have an ..."
Corrected

Section 2.1.2 within Barmet et al. the OH concentration needs to be determine with reference to another VOC with a known rate constant. For instance:
What was the other VOC chosen for comparison?

d9-butanol

Line 29 page 9: "evolvement" should be evolution
Corrected

Lines 30 – 31 page 10: is it reasonable that OH chemistry is not allowed to occur during dark aging? If there is ozonolysis of alkenes taking place, then there will be OH radicals produced. Even if it is a small pathway it shouldn't be ruled out unless there was negligible dark aging in the presence of $O_3$ alone (without $NO_3$ chemistry taking place), or if you could model NO3 production to show there is negligible ozonolysis occurring.

Only minor or no decrease of butanol-d9 was observed during dark aging which indicates that OH-radicals were not produced to any significant extent.

Line 15 Page 11: I believe it should say "physical properties" because I don't understand how physical reactions would be what is meant.
Sentence was reformulated as:
"Evolution of physical properties of particles and many chemical reactions involve more than just one compound, particles of the same size, or phase state, but there are exceptions. As an example, particles of the same size can coagulate during evolution and form larger particles."

Line 32 Page 11 / Line 1 Page 12: should read " which assessed the measurement error..." omit was.
Corrected

Line 8 page 12: replace "much" with many
Corrected

Line 8 page 12: omit the sentence starting with "Additionally" either talk about how it was interesting or remove the sentence.
Removed

Line 10/11 page 12: what is "correct structure"?
Correct structure = correct information of dependencies.
We added a sentence to chapter 2.3. of the revised manuscript after first sentence:
"As causal discovery algorithm attempts to search dependencies based on data, there might be incorrect dependencies in the structure formed by an algorithm."
and to the third sentence

"… dataset, but in a situation where we know the correct structure resulting the aging process".

Line 18 page 12: "which" should be replaced with 'whose'
Corrected

Line 20 page 12: what are Mass Action Kinetics system?
This was changed to Laws of Mass Action. In addition, references and the name of used R-package added. We found that episode-package has been removed from CRAN on December 2019. The code was modified to take this into account (download the package from Github instead of CRAN). Sentence was reformulated as

"In smaller datasets, differential equations are following the Laws of Mass Action, applied in R-package *episode* (Mikkelsen, 2017; Seinfeld and Pandis, 2016)."

Line 31 page 12: RMSE is not defined. (I presume root mean square error, and will assume that is what is meant)
RMSE is indeed Root Mean Square Error. This was already defined in chapter 2.5.4 in original manuscript (2.2.4 in revised manuscript).

Lines 33 – 34 page 12: Why is this weighting required? Why would some time series not have a standard deviation (standard deviation of what?)? This is not explained adequately.
Weighting was used to better equalize effects of each time series in RMSE. If weighting is not done, time series with larger absolute values would generally have larger effect on RMSE. The sentence is now reformulated, to clarify that each time series was divided with its own standard deviation.
"To equally weight each time series when calculating RMSE, each time series were scaled by dividing those with its standard deviation before calculating RMSE. In further text, we refer to this scaled version as RMSE."

Lines 3-4 page 13: Why is only 30% of the simulated dataset predicted?
30% is the length of the data used for evaluate prediction accuracy. So 100 simulated states were used to fit a model, and then the model was predicting the next 30. Sentence was reformulated as
"Prediction length was 30% of the simulated dataset used to fit a model."

Section 3.1

Lines 24 – 31 page 14: The problem I have with this paragraph is its order. 1) you show the measurement of error in the Tables. 2) there is good agreement! 3) This isn't surprising. 4) discuss the actual data.

I would suggest rearranging the content so you first say this table shows what you want to show. Then discuss those results, then talk about the goodness of agreement and the overall conclusion.
Corrected

Tables 3-5 and not intuitive and are not adequately described. For instance, what is "unc_fraction"? or what is nObs? This information is not included in the Table headings. Maybe the tables should be
adapted so they don't appear as if they were just directly taken from a code output. For example: corMean.mean does not convey helpful information. Purely call the column title "Correlation Mean"

For each table headings, the following has been added:
"Variable nObs is a number of observations in each time series. Variable unc_frac is a standard deviation of random noise added to each time series (sd of each time series is adjusted to 1). "

For tables 6 and 7, unc_frac 0.1 was used. This is added to the table headings.

corMean.mean is mean of the mean correlation in each replication (that's why there was double mean.

Text ".mean" in every table column heading is now replaced with overline.

Line 31 page 15, what is fittingness?
Changed to goodness of fit. Additionally, added "structure" before accuracy.

Figure 5: are there not background measurements? The Figures should all start at 0 on the y-axis. Why do the figures not start at 0 mass loading? What is going on before the initiating of dark aging, are the concentrations stable? Why is the agreement so poor for 3B SOA 1?

Chamber background aerosol concentrations were below 0.3 ug/m3 measured using AMS (Tiitta et al., (2016), Fig. 2). Only very little aerosol off-gassing from the chamber walls is observed (Leskinen et al., 2015). PTR background measurements are listed in Hartikainen et al., (2018), Table S6.

Figures 6 and 8 are now corrected so that y-axis starts from 0.

Before each experiment, there has been a stabilization period. This is mentioned in the first paragraph of chapter 2.

Starting point of Figure 6 is when the chamber is filled and stabilized, thus the concentrations are not zero. In order to clarify this, we added a sentence to the first paragraph of chapter 2 in the revised manuscript:

"The end of the stabilization period is considered as the starting point of our analysis."

The poor agreement for 3B SOA1 is probably due to the fact that 2B and 3B were both used to find dependence structure for the experiments. Formation process of SOA1 seem to be somewhat different in the two experiments. Our model has not been able to find such a structure that could represent the evolution of SOA1 in both.

Lines 13-14 page 16, why would $NO_3$ radicals change SOA1 to SOA2? If nitrate is important in this process, then it would be condensed phase process where N2O5 could be uptaken into the particle phase.

NO3 refers here particle-phase measured concentrations. This is now clarified in figure captions and in line 28 page 4:
"The AMS was used to characterize the chemical signatures of particulate chemical species (Org, NO3, SO4, NH4, and Chl) of which organic species as PMF factors (details below) and NO3 were used in this study."

Gas-phase NO3 has to be modeled to answer this question, which is out of scope of this study, in for example (Geyer et al., 2001) the necessary reactions are compiled. It is true that N2O5 and subsequent reaction with water is the major sink for NO3 radicals, i.e. NOx, but we can expect some influence of NO3 on SOA formation.

Based on PMF2 modelling SOA1 didn't change to SOA2 (see Tiitta et al., 2016, Fig. 5) but SOA1 concentration was quite stable after relatively fast SOA1 formation process. The species in the SOA1 are expected to be formed via ozonolysis of unsaturated compounds such as alkenes, dienes and terpenes.

It might also be that SOA1 is correlated to some other variable that is affecting SOA2. As causal discovery algorithm is based on observed dependencies, all connections might not be causal.

Line 14 page 16: why is NO2 attributed here and not NO3 radicals? Because in the gas-phase that is when NO3 chemistry would be important.

Based on Tiitta et al. (2016), the formation of secondary organic nitrate factor occur via two channels: one is through the NO3 radical oxidation in case of excessive NOx and O3 in the dark aging experiments (as described in previous chapter) and the another channel is through photochemistry via reactions of peroxy radical (RO2/ with NO (Atkinson, 2000)). This most likely explains why we are seeing the formation of SOA2 also after UV lights were switched on in presence of high NO in the experiment 4B (Fig. 5c, Tiitta et al., 2016)."

Hence, in UV ageing NO should be positively correlated with SOA2. In dark ageing, the higher the initial NO2 concentration the more SOA2 we should obtain, however, considering the timeline of one individual (dark) ageing experiment, SOA2 and NO2 should be inversely correlated because NO2 is consumed to produce SOA2 (and even more consumed to particulate nitrate).

References

Atkinson, R.: Atmospheric chemistry of VOCs and NO(x), Atmos. Environ., 34(12–14), 2063–2101, doi:10.1016/S1352-2310(99)00460-4, 2000.

Geyer, A., Alicke, B., Konrad, S., Schmitz, T., Stutz, J. and Platt, U.: Chemistry and oxidation capacity of the nitrate radical in the continental boundary layer near Berlin, J. Geophys. Res. Atmos., 106(D8), 8013–8025, doi:10.1029/2000JD900681, 2001.

Hartikainen, A., Yli-Pirilä, P., Tiitta, P., Leskinen, A., Kortelainen, M., Orasche, J., Schnelle-Kreis, J., Lehtinen, K. E. J., Zimmermann, R., Jokiniemi, J. and Sippula, O.: Volatile Organic Compounds from Logwood Combustion: Emissions and Transformation under Dark and Photochemical Aging Conditions in a Smog Chamber, Environ. Sci. Technol., 52(8), 4979–4988, doi:10.1021/acs.est.7b06269, 2018.

Hartikainen, A., Tiitta, P., Ihalainen, M., Yli-Pirilä, P., Orasche, J., Czech, H., Kortelainen, M., Lamberg, H., Suhonen, H., Koponen, H., Hao, L., Zimmermann, R., Jokiniemi, J., Tissari, J. and Sippula, O.: Photochemical transformation of residential wood combustion emissions: dependence of organic aerosol composition on OH exposure, Atmos. Chem. Phys., 20(11), 6357–6378, doi:10.5194/acp-20-6357-2020, 2020.

Isaacman-Vanwertz, G., Massoli, P., O'Brien, R., Lim, C., Franklin, J. P., Moss, J. A., Hunter, J. F., Nowak, J. B., Canagaratna, M. R., Misztal, P. K.,

Arata, C., Roscioli, J. R., Herndon, S. T., Onasch, T. B., Lambe, A. T., Jayne, J. T., Su, L., Knopf, D. A., Goldstein, A. H., Worsnop, D. R. and Kroll, J. H.: Chemical evolution of atmospheric organic carbon over multiple generations of oxidation, Nat. Chem., 10(4), 462–468, doi:10.1038/s41557-018-0002-2, 2018.

Leskinen, A., Yli-Pirilä, P., Kuuspalo, K., Sippula, O., Jalava, P., Hirvonen, M.-R., Jokiniemi, J., Virtanen, A., Komppula, M. and Lehtinen, K. E. J.: Characterization and testing of a new environmental chamber, Atmos. Meas. Tech., 8(6), 2267–2278, doi:10.5194/amt-8-2267-2015, 2015.

Mikkelsen, F. V.: episode: Estimation with Penalisation in Systems of Ordinary Differential Equations, [online] Available from: https://cran.r-project.org/package=episode, 2017.

Seinfeld, J. H. and Pandis, S. N.: Atmospheric Chemistry and Physics, edited by J. H. Seinfeld and S. N. Pandis, Wiley., 2016.

Tiitta, P., Leskinen, A., Hao, L., Yli-Pirilä, P., Kortelainen, M., Grigonyte, J., Tissari, J., Lamberg, H., Hartikainen, A., Kuuspalo, K., Kortelainen, A.-M., Virtanen, A., Lehtinen, K. E. J., Komppula, M., Pieber, S., Prévôt, A. S. H., Onasch, T. B., Worsnop, D. R., Czech, H., Zimmermann, R., Jokiniemi, J. and Sippula, O.: Transformation of logwood combustion emissions in a smog chamber: formation of secondary organic aerosol and changes in the primary organic aerosol upon daytime and nighttime aging, Atmos. Chem. Phys., 16(20), 13251–13269, doi:10.5194/acp-16-13251-2016, 2016.